# Amortized Projection Optimization for Sliced Wasserstein Generative Models

**Khai Nguyen**
Department of Statistics and Data Sciences
The University of Texas at Austin
Austin, TX 78712
khainb@utexas.edu

**Nhat Ho**
Department of Statistics and Data Sciences
The University of Texas at Austin
Austin, TX 78712
minhnhat@utexas.edu

## Abstract

Seeking informative projecting directions has been an important task in utilizing sliced Wasserstein distance in applications. However, finding these directions usually requires an iterative optimization procedure over the space of projecting directions, which is computationally expensive. Moreover, the computational issue is even more severe in deep learning applications, where computing the distance between two mini-batch probability measures is repeated several times. This nested loop has been one of the main challenges that prevent the usage of sliced Wasserstein distances based on good projections in practice. To address this challenge, we propose to utilize the *learning-to-optimize* technique or *amortized optimization* to predict the informative direction of any given two mini-batch probability measures. To the best of our knowledge, this is the first work that bridges amortized optimization and sliced Wasserstein generative models. In particular, we derive linear amortized models, generalized linear amortized models, and non-linear amortized models which are corresponding to three types of novel mini-batch losses, named *amortized sliced Wasserstein*. We demonstrate the favorable performance of the proposed sliced losses in deep generative modeling on standard benchmark datasets [1].

## 1 Introduction

Generative modeling is one of the most important tasks in machine learning and data science. Leveraging the expressiveness of neural networks in parameterizing the model distribution, deep generative models such as GANs [17], VAEs [23], and diffusion models [19, 54], achieve a significant quality of sampling images. Despite differences in the way of modeling the model distribution, optimization objectives of training generative models can be written as minimizing a discrepancy $\mathcal{D}(\cdot, \cdot)$ between data distribution $\mu$ and the model distribution $\nu_\phi$ with $\phi \in \Phi$, parameter space of neural networks weights, namely, we solve for $\hat{\phi} \in \arg\min_{\phi \in \Phi} \mathcal{D}(\mu, \nu_\phi)$. For example, Kullback–Leibler divergence is used in VAEs and diffusion models, Jensen–Shannon divergence appears in GANs, and f-divergences are utilized in f-GANs [43]. Because of the complexity of the neural networks $\phi$, closed-form optimal solutions to these optimization problems are intractable. Therefore, gradient-based methods and their stochastic versions are widely used in practice to approximate these solutions.

Recently, optimal transport-based losses, which we denote as $\mathcal{D}(\cdot, \cdot)$, are utilized to train generative models due to their training stability, efficiency, and geometrically meaning. Examples of these models include Wasserstein GAN [3] with the dual form of Wasserstein-1 distance [46], and OT-GANs [14, 51] with the primal form of Wasserstein distance and with Sinkhorn divergence [8]

---

[1]Code for the paper is published at `https://github.com/UT-Austin-Data-Science-Group/AmortizedSW`.

36th Conference on Neural Information Processing Systems (NeurIPS 2022).

between mini-batch probability measures. Although these models considerably improve the generative performance, there have been remained certain problems. In particular, Wasserstein GAN is reported to fail to approximate the Wasserstein distance [55] while OT-GAN suffers from high computational complexity of Wasserstein distance: $\mathcal{O}(m^3 \log m)$ and its curse of dimensionality: the sample complexity of $\mathcal{O}(m^{-1/d})$ where $m$ is the number of supports of two mini-batch measures. The entropic regularization [8] had been proposed to improve the computational complexity of approximating optimal transport to $\mathcal{O}(m^2)$ [1, 30, 31, 29] and to remove the curse of dimensionality [34]. However practitioners usually choose to use the slicing (projecting version) of Wasserstein distance [57, 11, 25, 42] due to a fast computational complexity $\mathcal{O}(m \log m)$ and no curse of dimensionality $\mathcal{O}(m^{-1/2})$. The distance is known as sliced Wasserstein distance (SW) [4]. Sliced Wasserstein distance is defined as the expected one-dimensional Wasserstein distance between two projected measures over the uniform distribution over the unit sphere. Due to the intractability of the expectation, Monte Carlo samples from the uniform distribution over the unit sphere are used to approximate the distance. The number of samples is often called the number of projections and it is denoted as $L$.

From applications, practitioners observe that sliced Wasserstein distance requires a sufficiently large number of projections $L$ relative to the dimension of data to perform well [25, 11]. Increasing $L$ leads to a linear increase in computational time and memory. However, when data lie in a low dimensional manifold, several projections are redundant since they collapse projected measures to a Dirac-Delta measure at zero. There are some attempts to overcome that issue including sampling orthogonal directions [49] and mapping the data to a lower-dimensional space [11]. The most popular approach is to search for the direction that maximizes the projected distance, which is known as max-sliced Wasserstein distance (Max-SW) [10]. Nevertheless, in the context of deep generative models and deep learning in general, the optimization over the unit sphere requires iterative projected gradient descent methods that can be computationally expensive. In detail, each gradient-update of the model parameters (neural networks) requires an additional loop for optimization of Max-SW between two mini-batch probability measures. Therefore, we have two nested optimization loops: the global loop (optimizing model parameters) and the local loop (optimizing projection). These optimization loops can slow down the training considerably.

**Contribution.** To overcome the issue, we propose to leverage *learning to learn* techniques (*amortized optimization*) to predict the optimal solution of the local projection optimization. We bridge the literature on amortized optimization and optimal transport by designing amortized models to solve the iterative optimization procedure of finding optimal slices in the sliced Wasserstein generative model. To the best of our knowledge, this is the first time amortized optimization is used in sliced Wasserstein literature. In summary, our main contributions are two-fold:

1. First, we introduce a novel family of mini-batch sliced Wasserstein losses that utilize amortized models to yield informative projecting directions, named amortized sliced Wasserstein losses ($\mathcal{A}$-SW). We specify three types of amortized models: linear amortized, generalized linear amortized, and non-linear amortized models that are corresponding to three mini-batch losses: linear amortized sliced Wasserstein ($\mathcal{LA}$-SW), generalized linear amortized sliced Wasserstein ($\mathcal{GA}$-SW), and non-linear amortized sliced Wasserstein ($\mathcal{NA}$-SW). Moreover, we discuss some properties of $\mathcal{A}$-SW losses including metricity, complexities, and connection to mini-batch Max-SW.

2. We then introduce the application of $\mathcal{A}$-SW in generative modeling. Furthermore, we carry out extensive experiments on standard benchmark datasets including CIFAR10, CelebA, STL10, and CelebAHQ to demonstrate the favorable performance of $\mathcal{A}$-SW in learning generative models. Finally, we measure the computational speed and memory of $\mathcal{A}$-SW, mini-batch Max-SW, and mini-batch SW to show the efficiency of $\mathcal{A}$-SW.

**Organization.** The remainder of the paper is organized as follows. We first provide background about Wasserstein distance, sliced Wasserstein distance, max-sliced Wasserstein distance, and amortized optimization in Section 2. In Section 3, we propose amortized sliced Wasserstein distances and analyze some of their theoretical properties. The discussion on related works is given in Section 4. Section 5 contains the application of $\mathcal{A}$-SW to generative models, qualitative experimental results, and quantitative experimental results on standard benchmarks. In Section 6, we provide a conclusion. Finally, we defer the proofs of key results and extra materials to the Appendices.

**Notation.** For any $d \geq 2$, $\mathbb{S}^{d-1} := \{\theta \in \mathbb{R}^d \mid ||\theta||_2^2 = 1\}$ denotes the $d$ dimensional unit hyper-sphere in $\mathcal{L}_2$ norm, and $\mathcal{U}(\mathbb{S}^{d-1})$ is the uniform measure over $\mathbb{S}^{d-1}$. Moreover, $\delta$ denotes the Dirac delta function. For $p \geq 1$, $\mathcal{P}_p(\mathbb{R}^d)$ is the set of all probability measures on $\mathbb{R}^d$ that has finite $p$-moments. For $\mu, \nu \in \mathcal{P}_p(\mathbb{R}^d)$, $\Pi(\mu, \nu) := \{\pi \in \mathcal{P}_p(\mathbb{R}^d \times \mathbb{R}^d) \mid \int_{\mathbb{R}^d} \pi(x,y)dx = \nu, \int_{\mathbb{R}^d} \pi(x,y)dy = \mu\}$ is the set of transportation plans between $\mu$ and $\nu$. For $m \geq 1$, we denotes $\mu^{\otimes m}$ as the product measure which has the supports are the joint vector of $m$ random variables that follows $\mu$. For a vector $X \in \mathbb{R}^{dm}$, $X := (x_1, \ldots, x_m)$, $P_X$ denotes the empirical measures $\frac{1}{m} \sum_{i=1}^{m} \delta_{x_i}$. We denote $\theta\sharp\mu$ as the push-forward probability measure of $\mu$ through the function $T_\theta : \mathbb{R}^d \to \mathbb{R}$ where $T_\theta(x) = \theta^\top x$.

## 2 Background

In this section, we first review the definitions of the Wasserstein distance, the sliced Wasserstein distance, and the max-sliced Wasserstein distance. We then formulate generative models based on the max-sliced Wasserstein distances and review the amortized optimization problem and its application to the max-sliced Wasserstein generative models.

### 2.1 (Sliced)-Wasserstein Distances

We first define the Wasserstein-$p$ distance [56, 45] between two probability measures $\mu \in \mathcal{P}_p(\mathbb{R}^d)$ and $\nu \in \mathcal{P}_p(\mathbb{R}^d)$ as follows: $W_p(\mu, \nu) := \left( \inf_{\pi \in \Pi(\mu,\nu)} \int_{\mathbb{R}^d \times \mathbb{R}^d} ||x - y||_p^p d\pi(x,y) \right)^{\frac{1}{p}}$. When $d = 1$, the Wasserstein distance has a closed form which is $W_p(\mu, \nu) = (\int_0^1 |F_\mu^{-1}(z) - F_\nu^{-1}(z)|^p dz)^{1/p}$ where $F_\mu$ and $F_\nu$ are the cumulative distribution function (CDF) of $\mu$ and $\nu$ respectively.

To utilize this closed-form property of Wasserstein distance in one dimension and overcome the curse of dimensionality of Wasserstein distance in high dimension, the sliced Wasserstein distance [4] between $\mu$ and $\nu$ had been introduced and admitted the following formulation: $SW_p(\mu, \nu) := \left( \int_{\mathbb{S}^{d-1}} W_p^p(\theta\sharp\mu, \theta\sharp\nu)d\theta \right)^{\frac{1}{p}}$. For each $\theta \in \mathbb{S}^{d-1}$, $W_p^p(\theta\sharp\mu, \theta\sharp\nu)$ can be computed in linear time $\mathcal{O}(n \log n)$ where $n$ is the number of supports of $\mu$ and $\nu$. However, due to the integration over the unit sphere, the sliced Wasserstein distance does not have closed-form expression. To approximate the intractable expectation, Monte Carlo scheme is used, namely, we draw uniform samples $\theta_1, \ldots, \theta_L \sim \mathcal{U}(\mathbb{S}^{d-1})$ from the unit sphere and obtain the following approximation: $SW_p(\mu, \nu) \approx \left( \frac{1}{L} \sum_{i=1}^{L} W_p^p(\theta_i\sharp\mu, \theta_i\sharp\nu) \right)^{\frac{1}{p}}$. In practice, $L$ should be chosen to be sufficiently large compared to the dimension $d$. It is not appealing since the computational complexity of SW is linear with $L$. To reduce projection complexity, max-sliced Wasserstein (Max-SW) is introduced [10]. In particular, the max-sliced Wasserstein distance between $\mu$ and $\nu$ is given by:

$$\text{Max-SW}(\mu, \nu) := \max_{\theta \in \mathbb{S}^{d-1}} W_p(\theta\sharp\mu, \theta\sharp\nu). \tag{1}$$

To solve the optimization problem, a projected gradient descent procedure is used. We present a simple algorithm in Algorithm 1. In practice, practitioners often set a fixed number of gradient updates, e.g., $T = 100$.

### 2.2 Learning Generative Models with Max-Sliced Wasserstein and Amortized Optimization

We now provide an application of (sliced)-Wasserstein distances to generative models settings. The problem can be seen as the following optimization:

$$\min_{\phi \in \Phi} \mathcal{D}(\mu, \nu_\phi), \tag{2}$$

where $\mathcal{D}(\cdot, \cdot)$ can be Wasserstein distance or SW distance or Max-SW distance. Despite the recent progress on scaling up Wasserstein distance in terms of the size of supports of probability measures [1, 30], using the original form of Wasserstein distances is still not tractable in real training due to both the memory constraint and time constraint. In more detail, the number of training samples is often huge, e.g., one million, and the dimension of data is also huge ,e.g., ten thousand. Therefore, mini-batch losses based on Wasserstein distances have been proposed [12, 40, 41]. The corresponding population form of these losses between two probability measures $\mu$ and $\nu$ is:

$$\tilde{\mathcal{D}}(\mu, \nu) := \mathbb{E}_{X,Y \sim \mu^{\otimes m} \otimes \nu^{\otimes m}} \mathcal{D}(P_X, P_Y), \tag{3}$$

---

**Algorithm 1** Max-sliced Wasserstein distance

---

**Input:** Probability measures: $\mu, \nu$, learning rate $\eta$, max number of iterations $T$.
Initialize $\theta$
**while** $\theta$ not converge or reach $T$ **do**
    $\theta = \theta + \eta \cdot \nabla_\theta \mathrm{W}_p(\theta \sharp \mu, \theta \sharp \nu)$
    $\theta = \frac{\theta}{||\theta||_2}$
**end while**
**Return:** $\theta$

---

**Algorithm 2** Training generative models with mini-batch max-sliced Wasserstein loss

---

**Input:** Data probability measure $\mu$, model learning rate $\eta_1$, slice learning rate $\eta_2$, model maximum number of iterations $T_1$, slice maximum number of iterations $T_2$, number of mini-batches $k$ (is often set to 1).
Initialize $\phi$, the model probability measure $\nu_\phi$
**while** $\phi$ not converge or reach $T_1$ **do**
    $\nabla_\phi = 0$
    Sample $(X_1, Y_{\phi,1}), \ldots, (X_k, Y_{\phi,k}) \sim \mu^{\otimes m} \otimes \nu_\phi^{\otimes m}$
    **for** $i = 1$ to $k$ **do**
        **while** $\theta$ not converge or reach $T_2$ **do**
            $\theta = \theta + \eta_2 \cdot \nabla_\theta \mathrm{W}_p(\theta \sharp P_{X_i}, \theta \sharp P_{Y_{\phi,i}})$
            $\theta = \frac{\theta}{||\theta||_2}$
        **end while**
        $\nabla_\phi = \nabla_\phi + \frac{1}{k} \nabla_\phi \mathrm{W}_p(\theta \sharp P_{X_i}, \theta \sharp P_{Y_{\phi,i}})$
    **end for**
    $\phi = \phi - \eta_1 \cdot \nabla_\phi$
**end while**
**Return:** $\phi, \nu_\phi$

---

where $m \geq 1$ is the mini-batch size and $\mathcal{D}$ is a Wasserstein metric.

In the generative model context [17], a stochastic gradient of the parameters of interest is utilized to update these parameters, namely,

$$\nabla_\phi \tilde{\mathcal{D}}(\mu, \nu_\phi) \approx \frac{1}{k} \sum_{i=1}^{k} \nabla_\phi \mathcal{D}(P_{X_i}, P_{Y_{\phi_i}}), \tag{4}$$

where $k$ is the number of mini-batches (is often set to 1), and $(X_i, Y_{\phi_i})$ is i.i.d sample from $\mu^{\otimes m} \otimes \nu_\phi^{\otimes m}$. The exchangeability between derivatives and expectation, and unbiasedness of the stochastic gradient are proven in [13]. Mini-batch losses are not distances; however, we can derive mini-batch energy distances from them [51].

**Learning generative models via max-sliced Wasserstein:** As we mentioned in Section 2.1, the max-sliced Wasserstein distance can overcome the curse of dimensionality of the Wasserstein distance and the issues of Monte Carlo samplings in the sliced Wasserstein distance. Therefore, it is an appealing divergence for learning generative models. By replacing the Wasserstein metric in equation (3), we arrive at the following formulation of the mini-batch max-sliced Wasserstein loss, which is given by:

$$\text{m-Max-SW}(\mu, \nu) = \mathbb{E}_{X,Y \sim \mu^{\otimes m} \otimes \nu^{\otimes m}} \left[ \max_{\theta \in \mathbb{S}^{d-1}} \mathrm{W}_p(\theta \sharp P_X, \theta \sharp P_Y) \right]. \tag{5}$$

Here, we can observe that each pair of mini-batch contains its own optimization problem of finding the "max" slice. Placing this in the context of iterative training of generative models, we can foresee its expensive computation. For a better understanding, we present an algorithm for training generative models with mini-batch max-sliced Wasserstein in Algorithm 2. In practice, there are some modifications of training generative models with mini-batch Max-SW for dealing with unknown metric space [11]. We defer the details of these modifications in Appendix C.

**Amortized optimization:** A natural question appears: "How can we avoid the nested loop in mini-batch Max-SW due to several local optimization problems?". In this paper, we propose a practical

solution for this problem, which is known as *amortized optimization* [2]. In amortized optimization, instead of solving all optimization problems independently, an amortized model is trained to predict optimal solutions to all problems. We now state the adapted definition of amortized models based on that in [52, 2]:

**Definition 1** *For each context variable $x$ in the context space $\mathcal{X}$, $\theta^\star(x)$ is the solution of the optimization problem $\theta^\star(x) = \arg\min_{\theta \in \Theta} \mathcal{L}(\theta, x)$, where $\Theta$ is the solution space. A parametric function $f_\psi : \mathcal{X} \to \Theta$, where $\psi \in \Psi$, is called an amortized model if*

$$f_\psi(x) \approx \theta^\star(x), \quad \forall x \in \mathcal{X}. \tag{6}$$

*The amortized model is trained by the amortized optimization objective which is defined as:*

$$\min_{\psi \in \Psi} \mathbb{E}_{x \sim p(x)} \mathcal{L}(f_\psi(x), x), \tag{7}$$

*where $p(x)$ is a probability measure on $\mathcal{X}$ which measures the "importance" of optimization problems.*

The amortized model in Definition 1 is sometimes called a *fully* amortized model for a distinction with the other concept of *semi* amortized model [2]. The gap between the predicted solution and the optimal solution $\mathbb{E}_{x \sim p(x)} ||f_\psi(x) - \theta^\star(x)||_2$ is called the amortization gap. However, understanding this gap depends on specific configurations of the objective $\mathcal{L}(\cdot, x)$, such as convexity and smoothness, which are often non-trivial to obtain in practice.

## 3 Amortized Sliced Wasserstein

In this section, we discuss an application of amortized optimization to the mini-batch max-sliced Wasserstein. In particular, we first formulate the approach into a novel family of mini-batch losses, named *Amortized Sliced Wasserstein*. Each member of this family utilizes an amortized model for predicting informative slicing directions of mini-batch measures. We then propose several useful amortized models in practice, including the linear model, the generalized linear model, and the non-linear model.

### 3.1 Amortized Sliced Wasserstein and Amortized Models

We extend the definition of the mini-batch max-sliced Wasserstein in Equation (5) with the usage of an amortized model to obtain the amortized sliced Wasserstein as follows.

**Definition 2** *Let $p \geq 1$, $m \geq 1$, and $\mu, \nu$ are two probability measures in $\mathcal{P}(\mathbb{R}^d)$. Given an amortized model $f_\psi : \mathbb{R}^{dm} \times \mathbb{R}^{dm} \to \mathbb{S}^{d-1}$ where $\psi \in \Psi$, the* amortized sliced Wasserstein *between $\mu$ and $\nu$ is:*

$$\mathcal{A}\text{-}SW(\mu, \nu) := \max_{\psi \in \Psi} \mathbb{E}_{(X,Y) \sim \mu^{\otimes m} \otimes \nu^{\otimes m}} [W_p(f_\psi(X, Y) \sharp P_X, f_\psi(X, Y) \sharp P_Y)]. \tag{8}$$

From the definition, we can see that the amortized model maps each pair of mini-batches to the optimal projecting direction on the unit hypersphere between two corresponding mini-batch probability measures. We have the following result about the symmetry and positivity of $\mathcal{A}$-SW.

**Proposition 1** *The amortized sliced Wasserstein losses are positive and symmetric. However, they are not metrics since they do not satisfy the identity property, namely, $\mathcal{A}\text{-}SW(\mu, \nu) = 0 \iff \mu = \nu$.*

Proof of Proposition 1 is in Appendix A.1. Our next result indicates that we can upper bound the amortized sliced Wasserstein in terms of mini-batch max-sliced Wasserstein.

**Proposition 2** *Assume that the space $\Psi$ is a compact set and the function $f_\psi$ is continuous in terms of $\psi$. Then, the amortized sliced Wasserstein are lower-bounds of the mini-batch max-sliced Wasserstein (Equation 5), i.e., $\mathcal{A}\text{-}SW(\mu, \nu) \leq m\text{-}Max\text{-}SW(\mu, \nu)$ for all probability measures $\mu$ and $\nu$.*

Proof of Proposition 2 is in Appendix A.2.

**Parametric forms of the amortized model:** Now we define three types of amortized models that we will use in the experiments.

**Definition 3** *Given $X, Y \in \mathbb{R}^{dm}$, and the one-one "reshape" mapping $T : \mathbb{R}^{dm} \rightarrow \mathbb{R}^{d \times m}$, the* linear amortized model *is defined as:*

$$f_\psi(X, Y) := \frac{w_0 + T(X)w_1 + T(Y)w_2}{||w_0 + T(X)w_1 + T(Y)w_2||_2}, \tag{9}$$

*where $w_1, w_2 \in \mathbb{R}^m$, $w_0 \in \mathbb{R}^d$ and $\psi = (w_0, w_1, w_2)$.*

In Definition 3, the assumption is that the optimal projecting direction lies on the subspace that is spanned by the basis $\{x_1, \ldots, x_m, y_1, \ldots, y_m, w_0\}$ where $X = (x_1, \ldots, x_m)$ and $Y = (y_1, \ldots, y_m)$. The computational complexity of this function is $\mathcal{O}((2m+1)d)$ since those of the operators $T(X)w_1$ and $T(Y)w_2$ are $\mathcal{O}(md)$ while adding the bias $w_0$ costs an additional computational complexity $\mathcal{O}(d)$. The number of parameters in linear amortized model is $2m + d$.

To increase the expressiveness of the linear amortized model, we apply some (non-linear) mappings to the inputs $X$ and $Y$, which results in the generalized linear amortized model as follows.

**Definition 4** *Given $X, Y \in \mathbb{R}^{dm}$, and the one-one "reshape" mapping $T : \mathbb{R}^{dm} \rightarrow \mathbb{R}^{d \times m}$, the* generalized linear amortized model *is defined as:*

$$f_\psi(X, Y) := \frac{w_0 + T(g_{\psi_1}(X))w_1 + T(g_{\psi_1}(Y))w_2}{||w_0 + T(g_{\psi_1}(X))w_1 + T(g_{\psi_1}(Y))w_2||_2}, \tag{10}$$

*where $w_1, w_2 \in \mathbb{R}^m$, $w_0 \in \mathbb{R}^d$, $\psi_1 \in \Psi_1$, $g_{\psi_1} : \mathbb{R}^{dm} \rightarrow \mathbb{R}^{dm}$ and $\psi = (w_0, w_1, w_2, \psi_1)$.*

In Definition 4, the assumption is that the optimal projecting direction lies on the subspace that is spanned by the basis $\{x'_1, \ldots, x'_m, y'_1, \ldots, y'_m, w_0\}$ where $g_{\psi_1}(X) = (x'_1, \ldots, x'_m)$ and $g_{\psi_1}(Y) = (y'_1, \ldots, y'_m)$. To specify, we let $g_{\psi_1}(X) = (W_2\sigma(W_1x_1) + b_0, \ldots, W_2\sigma(W_1x_m) + b_0)$, where $\sigma(\cdot)$ is the Sigmoid function, $W_1 \in \mathbb{R}^{d \times d}$, $W_2 \in \mathbb{R}^{d \times d}$, and $b_0 \in \mathbb{R}^d$. Compared to the linear model, the generalized linear model needs additional computations for $g_\psi(T(X))$ and $g_\psi(T(Y))$, which are at the order of $\mathcal{O}(2m(d^2 + d))$. It is because we need to include the complexity for matrix multiplication, e.g., $W_1x_1$ that costs $\mathcal{O}(d^2)$, for Sigmoid function that costs $\mathcal{O}(d)$, and for adding bias $b_0$ that costs $\mathcal{O}(d)$. Therefore, the total computational complexity of the function $f_\psi$ is $\mathcal{O}(4md^2 + 6md + d)$ while the number of parameters is $2(m + d^2 + d)$.

We finally propose another amortized model where we instead consider some mapping on the function $\omega_0 + T(X)\omega_1 + T(Y)\omega_2$ in the linear amortized model so as to increase the approximation power of the function $f_\psi$.

**Definition 5** *Given $X, Y \in \mathbb{R}^{dm}$, and the one-one "reshape" mapping $T : \mathbb{R}^{dm} \rightarrow \mathbb{R}^{d \times m}$, the* non-linear amortized model *is defined as:*

$$f_\psi(X, Y) := \frac{h_{\psi_2}(w_0 + T(X)w_1 + T(Y)w_2)}{||h_{\psi_2}(w_0 + T(X)w_1 + T(Y)w_2)||_2}, \tag{11}$$

*where $w_1, w_2 \in \mathbb{R}^m$, $w_0 \in \mathbb{R}^d$, $\psi_2 \in \Psi_2$, $h_{\psi_2} : \mathbb{R}^d \rightarrow \mathbb{R}^d$ and $\psi = (w_0, w_1, w_2, \psi_2)$.*

In Definition 5, the assumption is that the optimal projecting direction lies on the image of the function $h_{\psi_2}(\cdot)$ that maps from the subspace spanned by $\{x_1, \ldots, x_m, y_1, \ldots, y_m, w_0\}$ where $X = (x_1, \ldots, x_m)$ and $Y = (y_1, \ldots, y_m)$. The computational complexity for $h_{\psi_2}(x) = W_4\sigma(W_3x)) + b_0$ when $x \in \mathbb{R}^d$, $W_3 \in \mathbb{R}^{d \times d}$, $W_4 \in \mathbb{R}^{d \times d}$, and $b_0 \in \mathbb{R}^d$ is at the order of $\mathcal{O}(2(d^2 + d))$. Therefore, the total computational complexity of the function $f_\psi$ is $\mathcal{O}(2md + 2d^2 + 3d)$ while the number of parameters is $2(m + d^2 + d)$.

Using amortized models in Definitions 3-5 leads to three amortized sliced Wasserstein losses, which are linear amortized sliced Wasserstein loss ($\mathcal{L}\mathcal{A}$-SW), generalized linear amortized sliced Wasserstein loss ($\mathcal{G}\mathcal{A}$-SW), and non-linear amortized sliced Wasserstein loss ($\mathcal{N}\mathcal{A}$-SW) in turn.

**Remark 1** *The parametric forms in Definitions 3-5 are chosen as they are well-known choices for parametric functions. There are still several other ways of parameterization that can be utilized in practice based on prior knowledge about data, e.g., we can use convolution operator for saving parameters or we can strengthen the dependence between samples via recursive functions. We leave the design of these amortized models for future work.*

---

**Algorithm 3** Training generative models with amortized sliced Wasserstein loss

---

**Input:** Data probability measure $\mu$, model learning rate $\eta_1$, amortized learning rate $\eta_2$, maximum number of iterations $T$, number of mini-batches $k$ (is often set to 1).
Initialize $\phi$, the model probability measure $\nu_\phi$.
Initialize $\psi$, the amortized model $f_\psi$.
**while** $\phi, \psi$ not converge or reach $T$ **do**
$\quad \nabla_\phi = 0; \nabla_\psi = 0$
$\quad$ Sample $(X_1, Y_{\phi,1}), \ldots, (X_k, Y_{\phi,k}) \sim \mu^{\otimes m} \otimes \nu_\phi^{\otimes m}$
$\quad$ **for** $i = 1$ to $k$ **do**
$\quad\quad \nabla_\phi = \nabla_\phi + \frac{1}{k} \nabla_\phi \mathrm{W}_p(f_\psi(X_i, Y_{\phi,i}) \sharp P_{X_i}, f_\psi(X_i, Y_{\phi,i}) \sharp P_{Y_{\phi,i}})$
$\quad\quad \nabla_\psi = \nabla_\psi + \frac{1}{k} \nabla_\psi \mathrm{W}_p(f_\psi(X_i, Y_{\phi,i}) \sharp P_{X_i}, f_\psi(X_i, Y_{\phi,i}) \sharp P_{Y_{\phi,i}})$
$\quad$ **end for**
$\quad \phi = \phi - \eta_1 \cdot \nabla_\phi$
$\quad \psi = \psi + \eta_2 \cdot \nabla_\psi$
**end while**
**Return:** $\phi, \nu_\phi$

---

## 3.2 Amortized Sliced Wasserstein Generative Models

Based on the amortized sliced Wasserstein losses, our objective function for training a generative model $\nu_\phi$ parametrized by $\phi \in \Phi$ now becomes:

$$\min_{\phi \in \Phi} \max_{\psi \in \Psi} \mathbb{E}_{(X, Y_\phi) \sim \mu^{\otimes m} \otimes \nu_\phi^{\otimes m}} [\mathrm{W}_p(f_\psi(X, Y_\phi) \sharp P_X, f_\psi(X, Y_\phi) \sharp P_{Y_\phi})] := \min_{\phi \in \Phi} \max_{\psi \in \Psi} \mathcal{L}(\mu, \nu_\phi, \psi).$$

Since the above optimization forms a minimax problem, we can use an alternating stochastic gradient descent-ascent algorithm to solve it. In particular, the stochastic gradients of $\phi$ and $\psi$ can be estimated from mini-batches $(X_1, Y_{\phi,1}), \ldots, (X_k, Y_{\phi,k}) \sim \mu^{\otimes m} \otimes \nu_\phi^{\otimes m}$ as follows:

$$\nabla_\phi \mathcal{L}(\mu, \nu_\phi, \psi) = \frac{1}{k} \sum_{i=1}^{k} \nabla_\phi \mathrm{W}_p(f_\psi(X_i, Y_{\phi,i}) \sharp P_{X_i}, f_\psi(X_i, Y_{\phi,i}) \sharp P_{Y_{\phi,i}}), \tag{12}$$

$$\nabla_\psi \mathcal{L}(\mu, \nu_\phi, \psi) = \frac{1}{k} \sum_{i=1}^{k} \nabla_\psi \mathrm{W}_p(f_\psi(X_i, Y_{\phi,i}) \sharp P_{X_i}, f_\psi(X_i, Y_{\phi,i}) \sharp P_{Y_{\phi,i}}). \tag{13}$$

For more details, we present the procedure in Algorithm 3.

**Computational complexity:** From Algorithm 2 and Algorithm 3, we can see that training with $\mathcal{A}$-SW can escape the inner while-loop for finding the optimal projecting directions. In each iteration of the global while-loop, the computational complexity of computing the mini-batch Max-SW is $\mathcal{O}(2kT_2(m \log m + dm))$, which is composed by $k$ mini-batches with $T_2$ loops of the projection to one-dimension operator which costs $\mathcal{O}(2dm)$ and the computation of the sliced Wasserstein which costs $\mathcal{O}(2m \log m)$. For the mini-batch sliced Wasserstein, the overall computational complexity is $\mathcal{O}(2kL(m \log m + dm))$ where $L$ is the number of projections. For $\mathcal{L}\mathcal{A}$-SW, the overall computation complexity is $\mathcal{O}(2k(m \log m + 3md + d))$ where the extra complexity $\mathcal{O}((2m+1)d)$ comes from the computation of $f_\psi(\cdot)$ (see Section 3.1). Similarly, the computational complexities of $\mathcal{G}\mathcal{A}$-SW and $\mathcal{N}\mathcal{A}$-SW are respectively $\mathcal{O}(2k(m \log m + 4md^2 + 7md + d))$ and $\mathcal{O}(2k(m \log m + 3md + 2d^2 + 3d))$.

**Projection Complexity:** Compared to the sliced Wasserstein, Max-SW reduces the space for projecting directions from $\mathcal{O}(L)$ to $\mathcal{O}(1)$. For $\mathcal{L}\mathcal{A}$-SW, $\mathcal{G}\mathcal{A}$-SW, and $\mathcal{N}\mathcal{A}$-SW, the projection complexity is also $\mathcal{O}(1)$. However, compared to $d$ parameters of Max-SW, $\mathcal{L}\mathcal{A}$-SW needs $2m + d$ parameters for creating the projecting directions while $\mathcal{G}\mathcal{A}$-SW and $\mathcal{N}\mathcal{A}$-SW respectively need $\mathcal{O}(2(m + d^2 + d))$ parameters for producing the directions (see Section 3.1).

**Remark 2** *The computational complexities and the projection complexities of $\mathcal{G}\mathcal{A}$-SW and $\mathcal{N}\mathcal{A}$-SW are based on the specific parameterization that we choose in Section 3. We would like to recall that these complexities can be reduced by lighter parameterization as in the remark at the end of Section 3.1.*

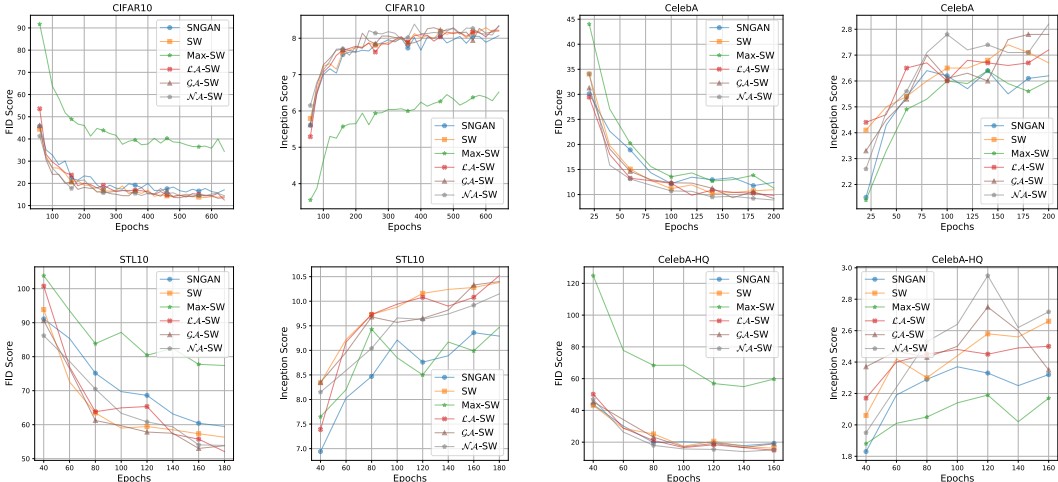

Figure 1: FID scores and IS scores over epochs of different training losses on datasets. We observe that members of $\mathcal{A}$-SW usually help the generative models converge faster.

## 4 Related Works

Generalized sliced Wasserstein [24] was introduced by changing the push-forward function from linear $T_\theta(x) = \theta^\top x$ to non-linear $T_\theta(x) = g(\theta, x)$ for some non-linear function $g(\cdot, \cdot)$. To cope with the projection complexity of sliced Wasserstein, a biased approximation based on the concentration of Gaussian projections was proposed in [37]. An implementation technique that utilizes both RAM and GPUs' memory for training sliced Wasserstein generative model was introduced in [27]. Augmenting the data to a higher-dimensional space for a better linear separation results in augmented sliced Wasserstein [6]. Projected Robust Wasserstein (PRW) metrics appeared in [44] that finds the best orthogonal linear projecting operator onto $d' > 1$ dimensional space. Riemannian optimization techniques for solving PRW were proposed in [28, 20]. We would like to recall that, amortized optimization techniques can be also applied to the case of PRW, max-K-sliced Wasserstein [9], sliced divergences [36], and might be applicable for sliced mutual information [16]. Statistical guarantees of training generative models with sliced Wasserstein were derived in [38].

Amortized optimization was first introduced in the form of amortized variational inference [23, 47]. Several techniques were proposed to improve the usage of amortized variational inference such as using meta sets in [58], using iterative amortized variational inference in [33], using regularization in [53]. Amortized inference was also applied into many applications such as probabilistic reasoning [15], probabilistic programming [48], and structural learning [5]. However, to the best of our knowledge, it is the first time that amortized optimization is used in the literature of optimal transport. We refer to [2] for a tutorial about the amortized optimization.

## 5 Experiments

In this section, we focus on comparing $\mathcal{A}$-SW generative models with SNGAN [35], the sliced Wasserstein generator [11], and the max-sliced Wasserstein generator [10]. The parameterization of model distribution is based on the neural network architecture of SNGAN [35]. The detail of the training processes of all models is given in Appendix C. For datasets, we choose standard benchmarks such as CIFAR10 (32x32) [26], STL10 (96x96) [7], CelebA (64x64), and CelebAHQ (128x128) [32]. For quantitative comparison, we use the FID score [18] and the Inception score (IS) [50]. We also show some randomly generated images from different models for qualitative comparison. We give full experimental results in Appendix D. The detailed settings about architectures, hyperparameters, and evaluation of FID and IS are given in Appendix E. We would like to recall that all losses that are used in this section are in their mini-batch version.

We first demonstrate the quality of using $\mathcal{A}$-SW in the training generative model compared to the baseline SNGAN, and other mini-batch sliced Wasserstein variants. Then, we investigate the convergence of generative models trained by different losses including the standard SNGAN's loss,

Table 1: Summary of FID and IS scores of methods on CIFAR10 (32x32), CelebA (64x64), STL10 (96x96), and CelebA-HQ (128x128). We observe that $\mathcal{A}$-SW losses provide the best results among all the training losses.

| Method | CIFAR10 (32x32) | | CelebA (64x64) | | STL10 (96x96) | | CelebA-HQ (128x128) | |
|---|---|---|---|---|---|---|---|---|
| | FID ($\downarrow$) | IS ($\uparrow$) | FID ($\downarrow$) | IS ($\uparrow$) | FID ($\downarrow$) | IS ($\uparrow$) | FID ($\downarrow$) | IS ($\uparrow$) |
| SNGAN | 17.09 | 8.07 | 12.41 | 2.61 | 59.48 | 9.29 | 19.25 | 2.32 |
| SW | 14.25$\pm$0.84 | 8.12$\pm$0.07 | 10.45 | 2.70 | 56.32 | 10.37 | 16.17 | 2.65 |
| Max-SW | 31.33$\pm$3.02 | 6.67$\pm$0.37 | 11.28 | 2.60 | 77.40 | 9.46 | 29.50 | 2.36 |
| $\mathcal{LA}$-SW (ours) | **13.21$\pm$0.69** | 8.19$\pm$0.03 | 9.82 | 2.72 | **52.08** | **10.52** | **14.94** | 2.50 |
| $\mathcal{GA}$-SW (ours) | 13.64$\pm$0.11 | 8.22$\pm$0.11 | 9.21 | 2.78 | 53.80 | 10.40 | 18.97 | 2.34 |
| $\mathcal{NA}$-SW (ours) | 14.22$\pm$0.51 | **8.29$\pm$0.08** | **8.91** | **2.82** | 53.90 | 10.14 | 15.17 | **2.72** |

mini-batch SW, mini-batch Max-SW, and $\mathcal{A}$-SW by looking at their FID scores and IS scores over training epochs of their best settings. After that, we compare models qualitatively by showing their randomly generated images. Finally, we report the training speed (number of training iterations per second) and the training memory (megabytes) of all settings of all training losses.

**Summary of FID and IS scores:** We show FID scores and IS scores of all models at the last training step on all datasets in Table 1. For SW and Max-SW, we select the best setting of hyperparameters for each score. In particular, we search for the best setting of the number of projections $L \in \{1, 100, 1000, 10000\}$. Also, we do a grid search on two hyperparameters of Max-SW, namely, the slice maximum number of iterations $T_2 \in \{1, 10, 100\}$ and the slice learning rate $\eta_2 \in \{0.001, 0.01, 0.1\}$. The detailed FID scores and IS scores for all settings are reported in Table 3 in Appendix D. For amortized models, we fix the slice learning rate $\eta_2 = 0.01$. From Table 1, the best amortized model provides lower FID scores and IS scores than SNGAN, SW, and Max-SW on all datasets of multiple image resolutions. We would like to recall that, SNGAN is reported to be better than WGAN [3] in [35]. Furthermore, the best generative models trained by $\mathcal{A}$-SW are better than models trained with SNGAN, SW, and Max-SW. Interestingly, the $\mathcal{LA}$-SW performs consistently well compared to other members of $\mathcal{A}$-SW. Also, we observe that Max-SW performs worse than both $\mathcal{A}$-SW and SW. This might be because the local optimization of Max-SW gets stuck at some bad optimum. However, we would like to recall that Max-SW is still better than SW with $L = 1$ (see Table 3 in Appendix D). It emphasizes the benefit of searching for a good direction for projecting.

**FID and IS scores over training epochs:** We show the values of FID scores and Inception scores over epochs on CIFAR10, CelebA, STL10, and CelebA-HQ in Figure 1. According to the figures in Figure 1, we observe that using SW and $\mathcal{A}$-SW helps the generative models converge faster than SNGAN. Moreover, FID lines of $\mathcal{A}$-SW are usually under the lines of other losses and the IS lines of $\mathcal{A}$-SW are usually above the lines of others. Therefore, $\mathcal{A}$-SW losses including $\mathcal{LA}$-SW, $\mathcal{GA}$-SW, and $\mathcal{NA}$-SW can improve the convergence of training generative models.

**Generated images:** We show generated images on CIFAR10, CelebA, STL10 from SNGAN, and $\mathcal{LA}$-SW in Figure 2 as a qualitative comparison. The generated images on CelebAHQ and the generated images of Max-SW, $\mathcal{GA}$-SW, and $\mathcal{NA}$-SW are given in Appendix D. From these images, we observe that the quality of generated images is consistent with the FID scores and the IS scores. Therefore, it reinforces the benefits of using $\mathcal{A}$-SW to train generative models. Again, we would like to recall that all generated images are completely random without cherry-picking.

**Computational time and memory:** We report the number of training iterations per second and the memory in megabytes (MB) in Table 2. We would like to recall that reported numbers are under some errors due to the state of the computational device. From the table, we see that $\mathcal{LA}$-SW is comparable to Max-SW and SW ($L = 1$) about the computational memory and the computational time. More importantly, $\mathcal{LA}$-SW is faster and consumes less memory than SW ($L \geq 100$) and Max-SW ($T_2 \geq 10$). Compared to SNGAN, SW variants increase the demand for memory and computation slightly. From $\mathcal{LA}$-SW to $\mathcal{GA}$-SW and $\mathcal{NA}$-SW, the computational time is slower slightly; however, we need between 800 to 2100 MB of memory in extra. Again, the additional memory depends on the chosen parameterization (see Section 3). From this table, we can see that using sliced Wasserstein models gives better generative quality than SNGAN but it also costs more computational time and memory. Among sliced Wasserstein variants, $\mathcal{LA}$-SW is the best option since it costs the least additional memory and time while it gives consistently good results. We refer to Section 3 for discussion of the time and projection complexities of $\mathcal{A}$-SW.

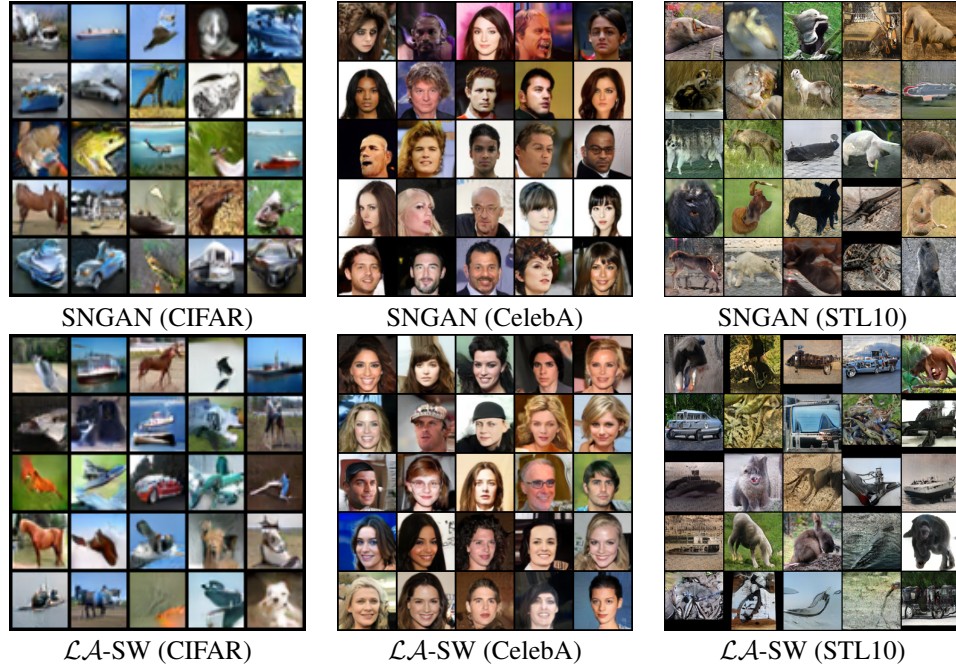

Figure 2: Random generated images of SNGAN and $\mathcal{LA}$-SW from CIFAR10, CelebA, and STL10.

Table 2: Computational time and memory of methods (in iterations per a second and megabytes (MB)).

| Method | CIFAR10 (32x32) | | CelebA (64x64) | | STL10 (96x96) | | CelebA-HQ (128x128) | |
|---|---|---|---|---|---|---|---|---|
| | Iters/s ($\uparrow$) | Mem ($\downarrow$) | Iters/s ($\uparrow$) | Mem ($\downarrow$) | Iters/s ($\uparrow$) | Mem ($\downarrow$) | Iters/s ($\uparrow$) | Mem ($\downarrow$) |
| SNGAN (baseline) | 19.97 | 1740 | 6.31 | 6713 | 9.33 | 3866 | 10.41 | 3459 |
| SW (L=1) | 18.73 | 2078 | 6.17 | 8011 | 9.31 | 4597 | 10.25 | 4111 |
| SW (L=100) | 18.42 | 2093 | 6.15 | 8015 | 9.11 | 4609 | 10.17 | 4120 |
| SW (L=1000) | 14.96 | 2112 | 6.13 | 8047 | 9.03 | 4616 | 9.63 | 4143 |
| SW (L=10000) | 5.84 | 2421 | 4.21 | 8353 | 6.50 | 4780 | 5.17 | 4428 |
| Max-SW ($T_2$=1) | 18.61 | 2078 | 6.17 | 8011 | 9.23 | 4597 | 10.22 | 4111 |
| Max-SW ($T_2$=10) | 18.16 | 2078 | 6.15 | 8011 | 9.17 | 4597 | 10.16 | 4111 |
| Max-SW ($T_2$=100) | 13.47 | 2078 | 5.78 | 8011 | 8.32 | 4597 | 8.13 | 4111 |
| $\mathcal{LA}$-SW (ours) | 18.58 | 2086 | 6.17 | 8021 | 9.23 | 4600 | 10.19 | 4115 |
| $\mathcal{GA}$-SW (ours) | 17.27 | 4151 | 6.07 | 10083 | 9.08 | 5251 | 10.11 | 6163 |
| $\mathcal{NA}$-SW (ours) | 17.67 | 4134 | 6.13 | 10068 | 9.11 | 5249 | 10.15 | 6152 |

## 6 Conclusion

We propose using amortized optimization for speeding up the training of generative models that are based on mini-batch sliced Wasserstein with projection optimization. We introduce three types of amortized models, including the linear, generalized, and non-linear amortized models, for predicting optimal projecting directions between all pairs of mini-batch probability measures. Moreover, using three types of amortized models leads to three corresponding mini-batch losses which are the linear amortized sliced Wasserstein, the generalized linear amortized sliced Wasserstein, and the non-linear amortized sliced Wasserstein. We then show that these losses can improve the result of training deep generative models in both training speed and generative performance.

## Acknowledgements

NH acknowledges support from the NSF IFML 2019844 and the NSF AI Institute for Foundations of Machine Learning.

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
