# Supplement to "Amortized Projection Optimization for Sliced Wasserstein Generative Models"

In this supplement, we first collect some proofs in Appendix A. We then introduce Amortized Projected Robust Wasserstein in Appendix B. Next, we discuss the training detail of generative models with different mini-batch losses in Appendix C. Moreover, we present detailed results on the deep generative model in Appendix D. Next, we report the experimental settings including neural network architectures, and hyper-parameter choices in Appendix E. Finally, we discuss the potential impacts of our works in Appendix F.

## A Proofs

In this appendix, we provide proofs for main results in the main text.

### A.1 Proof of Proposition 1

Recall that, the definition of $\mathcal{A}\text{-SW}(\mu,\nu)$ is as follows:
$$\mathcal{A}\text{-SW}(\mu,\nu) = \max_{\psi \in \Psi} \mathbb{E}_{(X,Y)\sim\mu^{\otimes m}\otimes\nu^{\otimes m}}[\mathrm{W}_p(f_\psi(X,Y)\sharp P_X, f_\psi(X,Y)\sharp P_Y)].$$

For the symmetric property of the amortized sliced Wasserstein, we have
$$\mathcal{A}\text{-SW}(\nu,\mu) = \max_{\psi \in \Psi} \mathbb{E}_{(Y,X)\sim\nu^{\otimes m}\otimes\mu^{\otimes m}}[\mathrm{W}_p(f_\psi(Y,X)\sharp P_X, f_\psi(Y,X)\sharp P_Y]$$
$$= \max_{\psi \in \Psi} \mathbb{E}_{(Y,X)\sim\nu^{\otimes m}\otimes\mu^{\otimes m}}[\mathrm{W}_p(f_\psi(X,Y)\sharp P_X, f_\psi(X,Y)\sharp P_Y]$$
$$= \max_{\psi \in \Psi} \mathbb{E}_{(X,Y)\sim\mu^{\otimes m}\otimes\nu^{\otimes m}}[\mathrm{W}_p(f_\psi(X,Y)\sharp P_X, f_\psi(X,Y)\sharp P_Y)]$$
$$= \mathcal{A}\text{-SW}(\mu,\nu),$$
where the second equality is because of the symmetry of Wasserstein distance, the third equality is due to the symmetry of $f_\psi(X,Y)$ (see forms of $f_\psi(X,Y)$ in Section 3). The positiveness of $\mathcal{A}\text{-SW}$ comes directly from the non-negativity of the Wasserstein distance.

To prove that $\mathcal{A}\text{-SW}$ violates the identity, we use a counter example where $\mu = \nu = \frac{1}{2}\delta_{x_1} + \frac{1}{2}\delta_{x_2}$ ($x_1 \neq x_2$). In this example, there exists a pair of mini-batches $X = (x_1, x_1)$ and $Y = (x_2, x_2)$. We choose $f_\psi(X,Y) = \frac{x_1+x_2}{||x_1+x_2||_2}$, then $f_\psi(X,Y)\sharp P_X \neq f_\psi(X,Y)\sharp P_Y$ which implies $\mathrm{W}_p(f_\psi(X,Y)\sharp P_X, f_\psi(X,Y)\sharp P_Y) > 0$. Since $\mathcal{A}\text{-SW}$ defines on the maximum value of $\psi \in \Psi$, $\mathcal{A}\text{-SW}(\mu,\nu) \geq \mathrm{W}_p(f_\psi(X,Y)\sharp P_X, f_\psi(X,Y)\sharp P_Y) > 0$.

### A.2 Proof of Proposition 2

Since the function $f_\psi$ is continuous in terms of $\psi$, it indicates that the function $\mathbb{E}_{(X,Y)\sim\mu^{\otimes m}\otimes\nu^{\otimes m}}[\mathrm{W}_p(f_\psi(X,Y)\sharp P_X, f_\psi(X,Y)\sharp P_Y)]$ is continuous in terms of $\psi$. Furthermore, as the parameter space $\Psi$ is compact, there exist $\psi^* \in \arg\max_{\psi\in\Psi} \mathbb{E}_{(X,Y)\sim\mu^{\otimes m}\otimes\nu^{\otimes m}}[\mathrm{W}_p(f_\psi(X,Y)\sharp P_X, f_\psi(X,Y)\sharp P_Y)]$. Then, we have
$$\mathcal{A}\text{-SW}(\mu,\nu) = \mathbb{E}_{(X,Y)\sim\mu^{\otimes m}\otimes\nu^{\otimes m}}[\mathrm{W}_p(f_{\psi^*}(X,Y)\sharp P_X, f_{\psi^*}(X,Y)\sharp P_Y)]$$
$$= \mathbb{E}_{(X,Y)\sim\mu^{\otimes m}\otimes\nu^{\otimes m}}[\mathrm{W}_p(\theta_{\psi^\star}\sharp P_X, \theta_{\psi^\star}\sharp P_Y)]$$
$$\leq \mathbb{E}_{(X,Y)\sim\mu^{\otimes m}\otimes\nu^{\otimes m}}\left[\max_{\theta\in\mathbb{S}^{d-1}} \mathrm{W}_p(\theta\sharp P_X, \theta\sharp P_Y)\right] := \text{m-Max-SW}(\mu,\nu).$$

As a consequence, we obtain the conclusion of the proposition.

## B Amortized Projected Robust Wasserstein

We first recall the definition of projected robust Wasserstein (PRW) distance [44]. Given two probability measures $\mu, \nu \in \mathcal{P}_p(\mathbb{R}^d)$, the projected robust Wasserstein distance between $\mu$ and $\nu$ is defined as:
$$PRW_k(\mu,\nu) := \max_{U\in\mathbb{V}_k(\mathbb{R}^d)} W_p(U\sharp\mu, U\sharp\nu), \tag{14}$$

Table 3: Summary of FID and IS scores of methods on CIFAR10 (32x32), CelebA (64x64), STL10 (96x96), and CelebA-HQ (128x128).

| Method | CIFAR10 (32x32) | | CelebA (64x64) | | STL10 (96x96) | | CelebA-HQ (128x128) | |
|---|---|---|---|---|---|---|---|---|
| | FID ($\downarrow$) | IS ($\uparrow$) | FID ($\downarrow$) | IS ($\uparrow$) | FID ($\downarrow$) | IS ($\uparrow$) | FID ($\downarrow$) | IS ($\uparrow$) |
| SNGAN (baseline) | 17.09 | 8.07 | 12.41 | 2.61 | 59.48 | 9.29 | 19.25 | 2.32 |
| SW (L=1) | 53.95 | 5.41 | 34.47 | 2.61 | 144.64 | 5.82 | 147.35 | 2.02 |
| SW (L=100) | 15.90±0.45 | 8.08±0.04 | 10.45 | 2.70 | 62.44 | 9.91 | 17.57 | 2.43 |
| SW (L=1000) | 14.58±0.95 | 8.10±0.06 | 10.96 | 2.67 | 57.12 | 10.25 | 16.17 | 2.65 |
| SW (L=10000) | 14.25±0.84 | 8.12±0.07 | 10.82 | 2.66 | 56.32 | 10.37 | 18.08 | 2.62 |
| Max-SW ($T_2$=1; $\eta_2$=0.001) | 35.52±1.97 | 6.54±0.22 | 11.28 | 2.60 | 101.37 | 7.98 | 34.97 | 1.98 |
| Max-SW ($T_2$=10; $\eta_2$=0.001) | 31.33±3.02 | 6.67±0.37 | 15.98 | 2.51 | 77.40 | 9.46 | 29.50 | 2.36 |
| Max-SW ($T_2$=100; $\eta_2$=0.001) | 41.20±2.33 | 6.02±0.25 | 16.52 | 2.46 | 86.91 | 9.05 | 56.20 | 2.26 |
| Max-SW ($T_2$=1; $\eta_2$=0.01) | 40.28±2.10 | 6.21±0.19 | 14.11 | 2.62 | 88.29 | 9.26 | 43.16 | 2.36 |
| Max-SW ($T_2$=10; $\eta_2$=0.01) | 39.56±4.55 | 6.25±0.36 | 16.89 | 2.49 | 90.82 | 9.18 | 59.74 | 2.16 |
| Max-SW ($T_2$=100; $\eta_2$=0.01) | 44.68±3.22 | 5.98±0.31 | 12.80 | 2.70 | 99.32 | 8.52 | 55.94 | 2.11 |
| Max-SW ($T_2$=1; $\eta_2$=0.1) | 36.60 | 6.58 | 18.87 | 2.42 | 94.33 | 8.19 | 52.68 | 2.16 |
| Max-SW ($T_2$=10; $\eta_2$=0.1) | 48.42 | 6.19 | 16.22 | 2.49 | 90.17 | 9.70 | 43.65 | 2.17 |
| Max-SW ($T_2$=100; $\eta_2$=0.1) | 50.74 | 5.42 | 14.40 | 2.59 | 101.38 | 8.46 | 42.81 | 2.20 |
| $\mathcal{L}\mathcal{A}$-SW (ours) | **13.21±0.69** | 8.19±0.03 | 9.82 | 2.72 | **52.08** | **10.52** | **14.94** | 2.50 |
| $\mathcal{G}\mathcal{A}$-SW (ours) | 13.64±0.11 | 8.22±0.11 | 9.21 | 2.78 | 53.80 | 10.40 | 18.97 | 2.34 |
| $\mathcal{N}\mathcal{A}$-SW (ours) | 14.22±0.51 | **8.29±0.08** | **8.91** | **2.82** | 53.90 | 10.14 | 15.17 | **2.72** |

where $\mathbb{V}_k(\mathbb{R}^d) := \{U \in \mathbb{R}^{d \times k} | U^\top U = I_k\}$ is the Stefel Manifold. PRW can be seen as the generalization of Max-SW since PRW with $k = 1$ is equivalent to Max-SW. Similar to Max-SW, the optimization of PRW is solved by using projected gradient ascent. The detailed of the algorithm is given in Algorithm 4. We would like to recall that other methods of optimization have also been used to solved PRW such as Riemannian optimization [28], block coordinate descent [21]. However, in this paper, we consider the original and simplest method which is projected gradient ascent.

In deep learning and large-scale applications, the mini-batch loss version of PRW is used, that is defined as follow:

$$\text{m-}PRW_k(\mu, \nu) = \mathbb{E}_{X, Y \sim \mu^{\otimes m} \otimes \nu^{\otimes m}} \left[ \max_{U \in \mathbb{V}_k(\mathbb{R}^d)} W_p(U \sharp P_X, U \sharp P_Y) \right]. \quad (15)$$

**Amortized Projected Robust Wasserstein loss:** We define Amortized Projected Rubust Wasserstein loss as follow:

**Definition 6** *Let $p \geq 1$, $m \geq 1$, and $\mu, \nu$ are two probability measures in $\mathcal{P}(\mathbb{R}^d)$. Given an amortized model $f_\psi : \mathbb{R}^{dm} \times \mathbb{R}^{dm} \to \mathbb{V}_k(\mathbb{R}^d)$ where $\psi \in \Psi$, the amortized projected robust Wasserstein between $\mu$ and $\nu$ is:*

$$\mathcal{A}\text{-}PRW(\mu, \nu) := \max_{\psi \in \Psi} \mathbb{E}_{(X,Y) \sim \mu^{\otimes m} \otimes \nu^{\otimes m}} [W_p(f_\psi(X, Y) \sharp P_X, f_\psi(X, Y) \sharp P_Y)]. \quad (16)$$

Similar to the case of $\mathcal{A}$-SW, $\mathcal{A}$-PRW is symmetric, positive, and is a lowerbound of PRW. Also, $\mathcal{A}$-PRW is not a metric since it does not satisfy the identity property.

**Amortized models:** Similar to the case of $\mathcal{A}$-SW, we can derive linear model, generalized linear model, and non-linear amortized model. The only change is that the model gives $k$ output vectors instead of 1 vector.

**Definition 7** *Given $X, Y \in \mathbb{R}^{dm}$, and the one-one "reshape" mapping $T : \mathbb{R}^{dm} \to \mathbb{R}^{d \times m}$, the linear projected amortized model is defined as:*

$$f_\psi(X, Y) := Proj_{\mathbb{V}_k(\mathbb{R}^d)}(W_0 + T(X)W_1 + T(Y)W_2), \quad (17)$$

*where $W_1, W_2 \in \mathbb{R}^{m \times k}, W_0 \in \mathbb{R}^{d \times k}$, and $Proj_{\mathbb{V}_k(\mathbb{R}^d)}$ return the Q matrix in QR decomposition.*

The definitions of the generalized linear projected amortized model and non-linear projected amortized model are straight-forward from the definitions of generalized linear model and non-linear model in $\mathcal{A}$-SW.

---

**Algorithm 4** Projected Robust Wasserstein distance

---

**Input:** Probability measures: $\mu, \nu$, learning rate $\eta$, max number of iterations $T$.
Initialize $U$
**while** $U$ not converge or reach $T$ **do**
    $U = U + \eta \cdot \nabla_U W_p(U \sharp \mu, U \sharp \nu)$
    $Q, R = QR(U)$ (QR decomposition)
    $U = Q$
**end while**
**Return:** $\theta$

---

## C   Training Generative Models

In this section, we review the parameterization of training losses of generative models.

**Parametrization:** We first discuss the parametrization of the model distribution $\nu_\phi$. In particular, $\nu_\phi$ is a pushforward probability measure that is created by pushing a unit multivariate Gaussian ($\epsilon$) through a neural network $G_\phi$ that maps from the realization of the noise to the data space. The detail of the architecture of $G_\phi$ is given in Appendix E. For training both SNGAN and generative models of SW, Max-SW, and $\mathcal{A}$-SW, we need a second neural network $T_\beta$ that maps from data space to a single scalar. The second neural network is called *Discriminator* in SNGAN or *Feature encoder* in the others. However, the architecture of the second neural network is the same for all models (see Appendix E). For the better distinction between training objectives of SNGAN and the objectives of the others, we denote $T_{\beta_1}$ is the sub neural network of $T_\beta$ that maps from the data space to a feature space (output of the last Resnet block), and $T_{\beta_2}$ that maps from the feature space (image of $T_{\beta_1}$) to a single scalar. More precisely, $T_\beta = T_{\beta_2} \circ T_{\beta_1}$. Again, we specify $T_{\beta_1}$ and $T_{\beta_1}$ in Appendix E.

**Training SNGAN:** Let $\mu$ is theta data probability measure, these two optimization problems are done alternatively in training SNGAN:

$$\min_{\beta_1, \beta_2} \left( \mathbb{E}_{x \sim \mu}[\min(0, -1 + T_{\beta_2}(T_{\beta_1}(x)))] + \mathbb{E}_{z \sim \epsilon}[\min(0, -1 - T_{\beta_2}(T_{\beta_1}(G_\phi(z))))] \right),$$

$$\min_\phi \mathbb{E}_{z \sim \epsilon}[-T_{\beta_2}(T_{\beta_1}(G_\phi(z)))].$$

**Training SW, Max-SW, and $\mathcal{A}$-SW:** For training these models, we adapt the framework in [11] to SNGAN, namely, we use these two objectives:

$$\min_{\beta_1, \beta_2} \left( \mathbb{E}_{x \sim \mu}[\min(0, -1 + T_{\beta_2}(T_{\beta_1}(x)))] + \mathbb{E}_{z \sim \epsilon}[\min(0, -1 - T_{\beta_2}(T_{\beta_1}(G_\phi(z))))] \right),$$

$$\min_\phi \tilde{\mathcal{D}}(\tilde{T}_{\beta_1, \beta_2} \sharp \mu, \tilde{T}_{\beta_1, \beta_2} \sharp G_\phi \sharp \epsilon),$$

where the function $\tilde{T}_{\beta_1, \beta_2} = [T_{\beta_1}(x), T_{\beta_2}(T_{\beta_1}(x))]$ which is the concatenation vector of $T_{\beta_1}(x)$ and $T_{\beta_2}(T_{\beta_1}(x))$, $\mathcal{D}$ is one of the mini-batch SW, the mini-batch Max-SW (see Equation 5), and $\mathcal{A}$-SW (see Definition 2). This technique is an application of metric learning since $\mathcal{L}_p$ norm is not meaningful on the space of natural images. This observation is mentioned in previous works [11, 14, 55, 39].

**Other settings:** The information about the mini-batch size, the learning rate, the optimizer, the number of iterations, and so on, are given in Appendix E.

## D   Full Experimental Results

**Detailed FID scores and Inception scores:** We first show the detailed FID scores and IS scores of all settings in Table 3. From the table, we can see that the quality of the SW depends on the number of projections. Namely, a higher number of projections often leads to better performance. For Max-SW, we obverse that increasing the number of iterations $T_2$ might not lead to a lower FID score and a higher IS score. The reason might be that the optimization gets stuck at some local optima. For the choice of the learning rate $\eta_2$, we do not see any superior setting for Max-SW.

**Generated Images:** We show generated images from SW, $\mathcal{G}\mathcal{A}$-SW, and $\mathcal{N}\mathcal{A}$-SW on CIFAR10, CelebA, and STL10 in Figure 3. The generated images from Max-SW on CIFAR10, CelebA, and

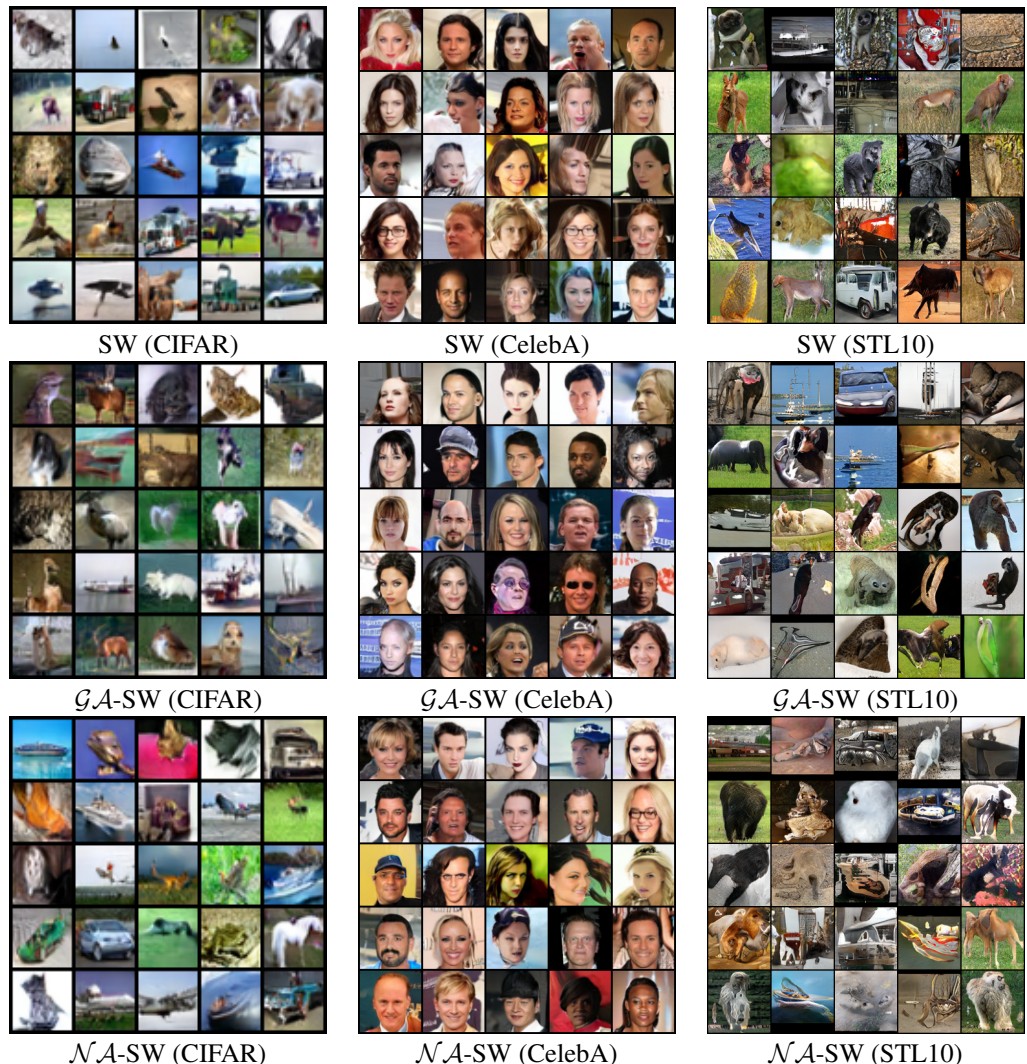

Figure 3: Random generated images of SW, $\mathcal{GA}$-SW, and $\mathcal{NA}$-SW from CIFAR10, CelebA, and STL10.

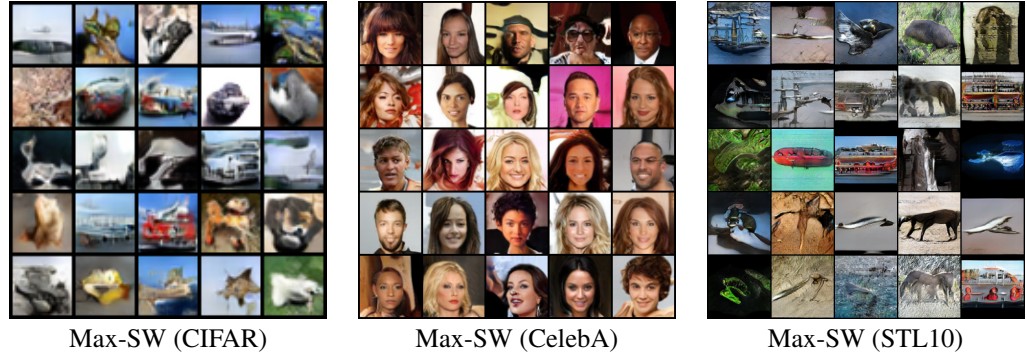

Figure 4: Random generated images of Max-SW from CIFAR10, CelebA, and STL10.

STL10 are given in Figure 4. The generated images from SNGAN and $\mathcal{LA}$-SW are given in Figure 5. The generated images from SW, Max-SW, $\mathcal{GA}$-SW, and $\mathcal{NA}$-SW on CelebA-HQ are presented in Figure 6. Again, we observe consistent quality results compared to the quantitative results of FID scores and Inception scores.

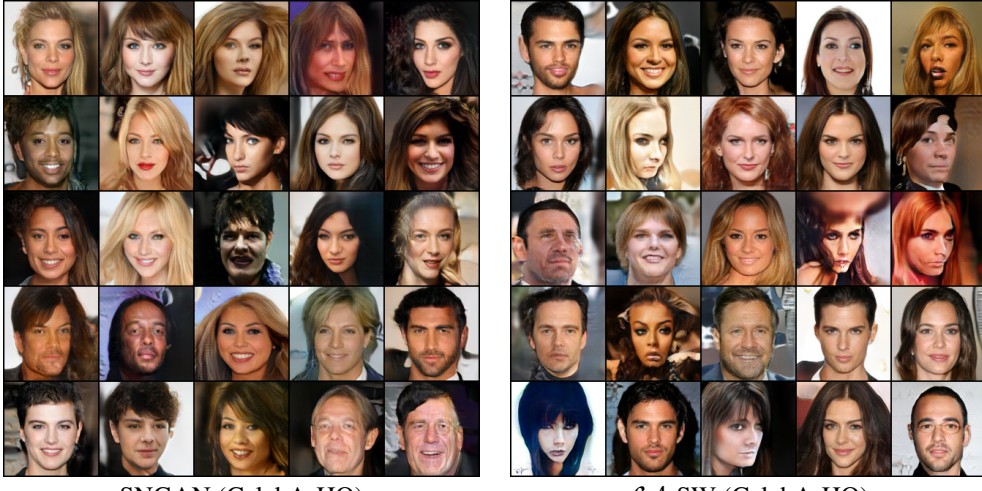

SNGAN (CelebA-HQ)  $\mathcal{LA}$-SW (CelebA-HQ)

Figure 5: Random generated images of SNGAN and $\mathcal{LA}$-SW from CelebAHQ.

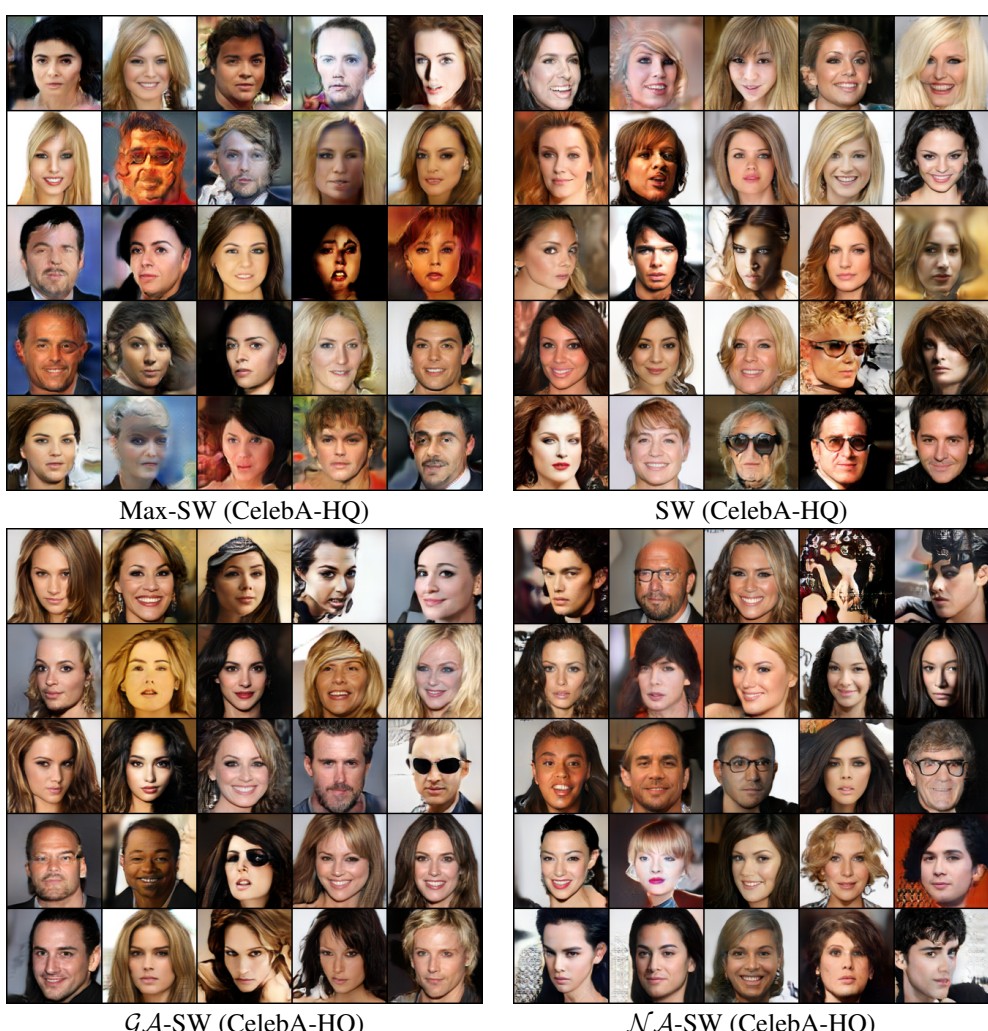

Max-SW (CelebA-HQ)  SW (CelebA-HQ)

$\mathcal{GA}$-SW (CelebA-HQ)  $\mathcal{NA}$-SW (CelebA-HQ)

Figure 6: Random generated images of Max-SW, SW, $\mathcal{GA}$-SW, and $\mathcal{NA}$-SW from CelebA-HQ.

**Results on Amortized PRW:** We present the result of training generative models on CIfAR10 with mini-batch PRW loss and amortized PRW losses in Table 4. For both PRW and $\mathcal{A}$-PRW,

Table 4: Summary of FID and IS scores of methods based on projected robust Wasserstein on CIFAR10 (32x32).

| Method | CIFAR10 (32x32) | |
| --- | --- | --- |
| | FID ($\downarrow$) | IS ($\uparrow$) |
| PRW (k=2) | 42.03 | 6.48 |
| $\mathcal{LA}$-PRW (k=2) (ours) | **14.27** | 8.02 |
| $\mathcal{GA}$-PRW (k=2) (ours) | 14.56 | 8.15 |
| $\mathcal{NA}$-PRW (k=2) (ours) | 14.69 | **8.43** |
| PRW (k=4) | 36.82 | 6.50 |
| $\mathcal{LA}$-PRW (k=4) (ours) | 14.33 | 8.01 |
| $\mathcal{GA}$-PRW (k=4) (ours) | **13.84** | **8.18** |
| $\mathcal{NA}$-PRW (k=4) (ours) | 14.68 | 8.05 |
| PRW (k=16) | 56.74 | 5.41 |
| $\mathcal{LA}$-PRW (k=16) (ours) | **14.16** | **8.06** |
| $\mathcal{GA}$-PRW (k=16) (ours) | 26.57 | 7.31 |
| $\mathcal{NA}$-PRW (k=16) (ours) | - | - |

Table 5: CIFAR10 architectures.

| (a) $G_\phi$ |
| --- |
| Input: $\epsilon \in \mathbb{R}^{128} \sim \mathcal{N}(0,1)$ |
| $128 \to 4 \times 4 \times 256$, dense linear |
| ResBlock up 256 |
| ResBlock up 256 |
| ResBlock up 256 |
| BN, ReLU, $3 \times 3$ conv, 3 Tanh |

| (b) $T_{\beta_1}$ |
| --- |
| Input: $\boldsymbol{x} \in [-1,1]^{32 \times 32 \times 3}$ |
| ResBlock down 128 |
| ResBlock down 128 |
| ResBlock down 128 |
| ResBlock 128 |
| ResBlock 128 |

| (c) $T_{\beta_2}$ |
| --- |
| Input: $\boldsymbol{x} \in \mathbb{R}^{128 \times 8 \times 8}$ |
| ReLU |
| Global sum pooling |
| $128 \to 1$ Spectral normalization |

we set the learning rate for $U$ is 0.01. We choose the best result from PRW with the number of gradient updates in $\{10, 100\}$ while we only update the amortized model once for $\mathcal{A}$-PRW. We observe that $\mathcal{A}$-PRW gives better FID and IS than PRW for all choice of $k \in \{2, 4, 16\}$. Moreover, linear amortized projected model gives the best result among amortized models. When $k = 16$, the non-linear amortized model suffers from numerical error when using QR decomposition, hence, we cannot provide the result for it. Overall, the result on PRW strengthen the claim that using amortized optimization for deep generative models with (sliced) projected Wasserstein can improve the result.

# E  Experimental Settings

**Neural network architectures:** We present the neural network architectures on CIFAR10 in Table 5, CelebA in Table 6, STL10 in Table 7, and CelebA-HQ in Table 8. In summary, we use directly the architectures from `https://github.com/GongXinyuu/sngan.pytorch`.

**Hyper-parameters:** For CIFAR10, CelebA, and CelebA-HQ, we set the training iterations to 50000 while we set it to 100000 in STL10. We update $T_{\beta_1}$ and $T_{\beta_2}$ every iterations while we update $G_\phi$ each 5 iterations. The mini-batch size $m$ is set to 128 on CIFAR10 and CelebA, is set to 32 on STL10, is set to 16 on CelebA-HQ. The learning rate of $G_\phi$, $T_{\beta_1}$, and $T_{\beta_2}$ is set to 0.0002. The optimizers for all optimization problems are Adam [22] with $(\beta_1, \beta_2) = (0, 0.9)$.

**FID scores and Inception scores:** For these two scores, we calculate them based on 50000 random samples from trained models. For FID scores, the statistics of datasets are calculated on all training samples.

Table 6: CelebA architectures.

**(a) $G_\phi$**

| Input: $\boldsymbol{\epsilon} \in \mathbb{R}^{128} \sim \mathcal{N}(0,1)$ |
| --- |
| $128 \to 4 \times 4 \times 256$, dense
linear |
| ResBlock up 256 |
| ResBlock up 256 |
| ResBlock up 256 |
| ResBlock up 256 |
| ResBlock up 256 |
| BN, ReLU,
$3 \times 3$ conv, 3 Tanh |

**(b) $T_{\beta_1}$**

| Input: $\boldsymbol{x} \in [-1,1]^{64 \times 64 \times 3}$ |
| --- |
| ResBlock down 128 |
| ResBlock down 128 |
| ResBlock down 128 |
| ResBlock 128 |
| ResBlock 128 |
| ResBlock 128 |

**(c) $T_{\beta_2}$**

| Input: $\boldsymbol{x} \in \mathbb{R}^{128 \times 8 \times 8}$ |
| --- |
| ReLU |
| Global sum pooling |
| $128 \to 1$
Spectral normalization |

Table 7: STL10 archtectures.

**(a) $G_\phi$**

| Input: $\boldsymbol{\epsilon} \in \mathbb{R}^{128} \sim \mathcal{N}(0,1)$ |
| --- |
| $128 \to 3 \times 3 \times 256$, dense
, linear |
| ResBlock up 256 |
| ResBlock up 256 |
| ResBlock up 256 |
| ResBlock up 256 |
| ResBlock up 256 |
| BN, ReLU,
$3 \times 3$ conv, 3 Tanh |

**(b) $T_{\beta_1}$**

| Input: $\boldsymbol{x} \in [-1,1]^{96 \times 96 \times 3}$ |
| --- |
| ResBlock down 128 |
| ResBlock down 128 |
| ResBlock down 128 |
| ResBlock down 128 |
| ResBlock 128 |
| ResBlock 128 |
| ResBlock 128 |

**(c) $T_{\beta_2}$**

| Input: $\boldsymbol{x} \in \mathbb{R}^{128 \times 6 \times 6}$ |
| --- |
| ReLU |
| Global sum pooling |
| $128 \to 1$
Spectral normalization |

Table 8: CelebA-HQ archtectures.

**(a) $G_\phi$**

| Input: $\boldsymbol{\epsilon} \in \mathbb{R}^{128} \sim \mathcal{N}(0,1)$ |
| --- |
| $128 \to 4 \times 4 \times 256$, dense
, linear |
| ResBlock up 256 |
| ResBlock up 256 |
| ResBlock up 256 |
| ResBlock up 256 |
| ResBlock up 256 |
| BN, ReLU,
$3 \times 3$ conv, 3 Tanh |

**(b) $T_{\beta_1}$**

| Input: $\boldsymbol{x} \in [-1,1]^{128 \times 128 \times 3}$ |
| --- |
| ResBlock down 128 |
| ResBlock down 128 |
| ResBlock down 128 |
| ResBlock down 128 |
| ResBlock 128 |
| ResBlock 128 |
| ResBlock 128 |

**(b) $T_{\beta_2}$**

| Input: $\boldsymbol{x} \in \mathbb{R}^{128 \times 8 \times 8}$ |
| --- |
| ReLU |
| Global sum pooling |
| $128 \to 1$
Spectral normalization |

# F  Potential Impact and Limitations

**Potential Impact:**  This work improves training generative models with sliced Wasserstein by using amortized optimization. Moreover, amortized sliced Wasserstein losses can be applied to various

applications such as generative models, domain adaptation, and approximate inference, adversarial attack, and so on. Due to its widely used potential, it can be used as a component in some applications that do not have a good purpose. For example, some examples are creating images of people without permission, attacking machine learning systems, and so on.

**Limitations:** In the paper, we have not been able to investigate the amortization gaps of the proposed amortized models since the connection of the optima of Max-SW to the supports of two probability measures has not been well-understand yet. Moreover, the design of amortized models requires more engineering to achieve better performance since there is no inductive bias for designing them at the moment. The hardness in designing amortized models is that we need to trade-off between the performance and computational efficiency. We will leave these questions to future work.