# OpenReview forum: "Amortized Projection Optimization for Sliced Wasserstein Generative Models"
_NeurIPS.cc/2022/Conference — NeurIPS 2022 Accept_

### Official Review · Reviewer_hTkP · 2022-07-01

**Rating:** 4
**Confidence:** 5
**Soundness:** 1 poor
**Presentation:** 3 good
**Contribution:** 2 fair

**Summary:**

Modern generative models aim at fitting the probability distribution of the observed data. To this end, a common strategy consists in minimizing a chosen divergence between the observed empirical distribution and a parametric distribution, over the set of parameters. In recent years, models based on the minimization of optimal transport (OT) metrics have attracted significant interest. However, the standard OT metric, namely the Wasserstein distance, suffers from an expensive computational cost and sample complexity in large-scale settings (i.e. when the compared distributions are high-dimensional or supported on a large number of samples). For that reason, using practical alternatives to the traditional Wasserstein distance is more convenient in generative modeling.

Due to its lower computational complexity and favorable theoretical properties, the sliced Wasserstein distance has been successfully integrated within the generative modeling framework. Nevertheless, the sliced Wasserstein distance (SW) is defined as an expectation which is intractable in general. In practice, this distance is thus traditionally approximated with a simple Monte Carlo algorithm: the expectation is replaced by a finite sum over L projection directions. Computing the sliced Wasserstein distance then scales linearly with L, meaning that improving the accuracy of the Monte Carlo estimator (by increasing L) leads to higher computational requirements. This is a limitation of SW-based generative algorithms, since they need to compute SW at each training iteration. One solution is to pick "more informative" samples in the Monte Carlo strategy, which motivated the formulation of maximum sliced Wasserstein (max-SW): the expectation is replaced by a maximum operator.

In this paper, the authors argue that the mini-batch formulation of max-SW (mini-batch max-SW, equation (5)) can be computationally expensive when used in generative modeling, and they develop alternative pseudo-metrics inspired by amortized optimization to overcome this problem. Their main contributions are summarized below.

1) This paper introduces a new family of pseudo-metrics, coined "amortized sliced Wasserstein" (A-SW), by reformulating the mini-batch max-SW using amortized optimization (Definition 2). In other words, the optimal projection directions found in mini-batch max-SW are approximated with a family of parametric functions (the amortized model).

2) The authors prove that amortized SW are positive, symmetric (Proposition 1) and lower than mini-batch max-SW (Proposition 1).

3) Three instances of amortized SW are proposed by specifying the amortized model (Definitions 3, 4 and 5) and an analysis of their computational complexity as compared to mini-batch-max-SW is provided.

4) Finally, GANs based on A-SW are proposed and compared against SNGAN and (max-)SW-based generators in image generation in terms of training speed, memory and quality of results, on 4 standard datasets (Section 5). According to the empirical results, A-SW is able to produce images of higher quality, at the price of a higher execution time and memory due to the optimization of the amortized model (depending on its parameterization).

**Questions:**

To address my concerns, I encourage the authors to:
- Clarify the motivation and take-home message of the paper: A-SW does not seem to provide computational advantages over mini-batch max-SW but instead, provides more accurate results ; this might be due to the design of the amortized models, which constraints the shape of the optimal projections, thus regularizes the underlying optimization problem.
- Add a more extensive discussion on the amortization gap to better motivate the proposed amortized models and the training procedure.
- Further illustrate the empirical superior performance of A-SW by: 1) running the experiments multiple times and report the associated error bars (Table 1, Figure 1); 2) finding an experiment where the training of mini-batch-max-SW-generator does not get stuck in a bad local optimum, and compare its performance against A-SW. It would also be interesting to see how mini-batch max-SW performs when the number of mini batches k is equal to 1.

Typos:
Main document: l.92, l.263 (unclear sentences); l.102 (definition of $W_p$: the norm should be Euclidean so indexed by 2); equation (2) ($\nu_\phi$ instead of $\nu$); l.131 ($Y_{\phi,i}$ instead of $Y_{\phi_i}$); l.174 ("two probability measureS")

Appendix: derivations below l.523 (parenthesis issue, equation repeated twice); l.553; legend of Table 7.

**Limitations:**

The limitations of A-SW are discussed in Section 5 and seem to be the computational complexity (which, again, contradicts the original motivation). The potential negative societal impact is discussed briefly in Appendix F.

**Strengths And Weaknesses:**

Strengths:
- The problem addressed in this paper is adequate for NeurIPS: generative models based on optimal transport metrics, in particular variants of sliced Wasserstein distance, have gained popularity in the machine learning community, and it is important to develop strategies which further reduce their computational requirements.
- The main idea behind the paper is quite original since, to the best of my knowledge, amortized optimization (which has been previously deployed in other areas such as variational inference and auto-encoders, for example) had not been applied in optimal transport.
- Overall, I found the paper clearly written and easy to follow. Some typos are reported in "Questions".

Weaknesses:

Unclear conclusion: In the end, the message of this paper is confusing to me: in the introduction, the authors motivate the development of A-SW as a strategy to reduce the computational complexity of mini-batch max-SW, but this aspect is not supported by the computational complexity analysis (l.237-247) nor the experimental section (l.324-326: A-SW is "comparable to Max-SW ... about the computational memory and the computational time", or even slower and more memory-intensive when the amortized model is not linear (Table 2)). According to Table 1, the main advantage in using A-SW instead of mini-batch-max-SW in generative models is not the speeding-up of the training, but the improved quality of the results (Table 1). However, it is not clear why this happens, which leads me to the next point ("Lack of theoretical justification").

Lack of theoretical justification: the proposed methodology is not theoretically-grounded in my opinion and the empirical analysis is not comprehensive enough to compensate this lack of insights, which makes me question the performance of the proposed generative algorithm. More specifically,
1) Proof of Proposition 2 (Appendix A.2): The conditions guaranteeing the existence of $\phi^*$ should be specified.
2) The parameterization of the amortized models is not clearly motivated. Are the proposed parameterizations standard in amortized optimization (if so, please add some references)? In particular, the amortization gap seems to be an important aspect in amortized optimization and is usually taken into account in the final methodology [1,2], but the authors chose not to study it (l.162-163). Are there any guarantees on whether the optimal projection directions lie in the space parameterized by Definitions 3, 4 or 5?
3) A-SW outperforms mini-batch max-SW in terms of FID/IS scores in the image generation problem, and the authors conjecture that mini-batch max-SW performs poorly because it is stuck in a local optimum. What prevents A-SW from suffering from the same issue? This aspect could have also been explored empirically, which relates to my last point ("No uncertainty measure in the experiments").

No uncertainty measure in the experiments: In Section 5, if I am not mistaken, generative models have been trained only once on each dataset. In the checklist, the authors specified that they reported error bars, but I cannot find these anywhere. Having multiple runs is essential and can support the conjecture that A-SW-based generative models do not get stuck in a local optimum or at least less often than mini-batch max-SW generators (I am not fully convinced by this aspect for now).

[1] "Amortized Inference Regularization", Shu et al. (NeurIPS 2018)

[2] "Iterative Amortized Policy Optimization", Marino et al. (NeurIPS 2021)

---

> ### Author Response · Authors · 2022-08-01
> **Response to Reviewer hTkP - Part 1**
>
> **Q14**: Unclear conclusion: In the end, the message of this paper is confusing to me: in the introduction, the authors motivate the development of A-SW as a strategy to reduce the computational complexity of mini-batch max-SW, but this aspect is not supported by the computational complexity analysis (l.237-247) nor the experimental section (l.324-326: A-SW is "comparable to Max-SW ... about the computational memory and the computational time", or even slower and more memory-intensive when the amortized model is not linear (Table 2)). According to Table 1, the main advantage of using A-SW instead of mini-batch-max-SW in generative models is not the speeding-up of the training, but the improved quality of the results (Table 1). However, it is not clear why this happens, which leads me to the next point ("Lack of theoretical justification").
>
> **A14**: Thanks for your comment. $\mathcal{A}$-SW gives the computational complexity of $\mathcal{O}(n\log n)$  that is the same as those of SW and Max-SW, while the memory complexity is comparable to Max-SW as discussed in the paper. From Table 2, when Max-SW uses 10 update steps for finding the max projecting direction, the $\mathcal{LA}$-SW variant is faster than it. When Max-SW uses 100 update steps, it is slower than all $\mathcal{A}$-SW variants.  We would like to recall that the computational benefit of amortized optimization is that it does not start local optimization problems from the start. In particular, optimums are predicted from an amortized model, and those models are trained on all local problems. That is the reason why amortized optimization can save computational time.
>
> The reason why $\mathcal{A}$-SW gives better performance than Max-SW is that Max-SW might get stuck at a local optimum in the optimization of finding the max projecting direction. Since Max-SW uses  the  projected gradient ascent to solve its optimization problem, it is not guaranteed to converge to the global optima. $\mathcal{A}$-SW introduces an assumption to predict the global optima which is the (generalized) linear combination of supports of the two distributions. Hence, that prior might lead to a better landscape for the optimization problem. Moreover, the amortized model is shared across all mini-batch distributions that avoid starting the optimization from the beginning like in Max-SW. As a result, ASW can find better projections in each mini-batch than Max-SW.
>
>
> **Q15**: Proof of Proposition 2 (Appendix A.2): The conditions guaranteeing the existence of ϕ∗ should be specified.
>
> **A15**: In the revised version of Proposition 2, we already include the additional conditions that the space $\Psi$ is compact and the map $f_{\psi}$ is continuous in terms of $\psi$ to guarantee the existence of $\psi^{*}$. Please refer to Appendix A.2 for new proof.
>
>
> **Q16**: The parameterization of the amortized models is not clearly motivated. Are the proposed parameterizations standard in amortized optimization (if so, please add some references)?
>
> **A16**: Thanks for your question. Since this is the first work that connects amortized optimization to sliced Wasserstein literature, we believe that our parameterizations of amortized models are original.  Our proposed amortized models are motivated by famous literature on generalized linear models [R.5] and they are the most natural ways to choose in practice.
>
> For the linear model, the assumption is that the optimal projecting direction lies on the subspace that is spanned by the basis \{$x_1,\ldots,x_m,y_1,\ldots,y_m,w_0$\} where $X=(x_1,\ldots,x_m)$ and $Y=(y_1,\ldots,y_m) $ are supports of two mini-batch measures, and $w_0$ is the vector of biases of the linear model. Similarly, the generalized linear model assumes that the optimal projecting direction lies in the subspace \{$x_1’,\ldots,x_m’,y_1’,\ldots,y_m’,w_0$\} where $X’=(x_1’,\ldots,x_m’) = g_{\psi_1}(X)$ and $Y’=(y_1’,\ldots,y_m’)= g_{\psi_1}(Y)$ for some non-linear link function $ g_{\psi_1}(.)$. Moreover, we would like to recall that we can use several architectures of neural networks for the amortized model. In the paper, our non-linear amortized model can be seen as a two-layer MLP. However, a more powerful amortized model leads to higher computational complexity as noticed by the reviewer. In the paper, we recommend the linear amortized model which is the most efficient model.
>
> [R.5] Plane Answers to Complex Questions, Christensen et al

---

> ### Author Response · Authors · 2022-08-01
> **Response to Reviewer hTkP - Part 2**
>
> **Q17**: the amortization gap seems to be an important aspect of amortized optimization and is usually taken into account in the final methodology [1,2], but the authors chose not to study it (l.162-163). Are there any guarantees on whether the optimal projection directions lie in the space parameterized by Definitions 3, 4 or 5?
>
> **A17**: We would like to remark that the nature of the optimization problem in mini-batch Max-SW is different from the conventional ElBO in VAE literature; hence, it is non-trivial to understand the amortization gap at the moment. Moreover, since the optimal solution of Max-SW has not been investigated rigorously in the literature yet, it is non-trivial to derive guarantees on whether the optimal projection directions lie in the space parameterized by Definitions 3, 4 or 5.
>
>
> **Q18**: A-SW outperforms mini-batch max-SW in terms of FID/IS scores in the image generation problem, and the authors conjecture that mini-batch max-SW performs poorly because it is stuck in a local optimum. What prevents A-SW from suffering from the same issue? This aspect could have also been explored empirically, which relates to my last point ("No uncertainty measure in the experiments").
>
> **A18**: Due to limitations of time and hardware, we can only run min-batch SW (L=100,1000,10000), mini-batch Max-SW ($\eta_2 \in$ \{0.001,0.01\} and $T_2$ in \{1,10,100\}), $\mathcal{LA}$-SW, $\mathcal{GA}$-SW, and $\mathcal{NA}$-SW five different times on CIFAR10. We evaluate the FID scores and IS scores on only the last epoch to speed up the experiments, hence, we cannot plot the FID curve and IS curve. In the revision, we have updated the results with error bars in Table 1. According to the table, the relative comparisons of FID scores and IS scores are unchanged. Namely, $\mathcal{LA}$-SW and $\mathcal{NA}$-SW give the best FID score and IS score respectively on CIFAR10. Moreover, we observe that Max-SW still gives a not good result which suggests that the projected gradient ascent of Max-SW leads to non-optimal slicing projection.
>
> As answered in **A11**, one reason that A-SW can avoid the local optimum better than Max-SW is that A-SW uses an amortized model to predict the optimum. The amortized model is trained on multiple local optimization problems which are finding the max slice on all pairs of mini-batch distributions. Therefore, the shared information between mini-batches might be one reason that A-SW can yield better results. Moreover, we design the amortized model to predict the global optima as a (generalized) linear combination of supports of the two distributions that might lead to a better landscape for the optimization problem.
>
> We also would like to refer the reviewer to the new experiments where we compare mini-batch projected subspace robust (PRW) which is the generalization of mini-batch Max-SW to $\mathcal{A}$-PRW which is the generalization of $\mathcal{A}$-SW. In more detail, they project two original distributions to a subspace with dimension $k>1$. The detailed of PRW and $\mathcal{A}$-PRW are given in Appendix C. By modifying our proposed amortized model slightly, we can derive three variants of  $\mathcal{A}$-PRW including $\mathcal{LA}$-PRW, $\mathcal{GA}$-PRW, and $\mathcal{NA}$-PRW. We again conduct experiments on generative models to compare $\mathcal{A}$-PRW variants with PRW with $k\in $\{2,4,16\}  on CIFAR10 dataset. The result is given in Table 4 in Appendix E and the detailed settings are also given in Appendix E. From the table, we observe that $\mathcal{A}$-PRW variants give both lower FID scores and IS scores than PRW for all choices of $k$. This further strengthens the performance of amortized optimization to find the best projecting subspace and the best projecting directions.
>
>
> **Q19**: Finding an experiment where the training of mini-batch-max-SW-generator does not get stuck in a bad local optimum, and compare its performance against A-SW. It would also be interesting to see how mini-batch max-SW performs when the number of mini-batches k is equal to 1.
>
> **A19**: Due to the limitation of time and hardware, we have not been able to design such experiments. We will report the experiment as soon as it is available. According to the number of mini-batches $k$, it has been already set to 1 for all methods in our experiments.

---

> ### Author Response · Authors · 2022-08-01
> **Response to Reviewer hTkP - Part 3**
>
> **Q20**: Clarify the motivation and take-home message of the paper: A-SW does not seem to provide computational advantages over mini-batch max-SW but instead, provides more accurate results ; this might be due to the design of the amortized models, which constraints the shape of the optimal projections, thus regularizes the underlying optimization problem.
>
> **A20**: As answered in $A11$,  the main benefit of amortized optimization is avoiding starting local optimization from the start. Based on Table 2, when Max-SW uses a lot of update steps, e.g., 100, it is slower than $\mathcal{A}$-variants. The reason that we focus on the computational benefit instead of the performance is that the computational benefit of amortized optimization is well-known. The better performance of $\mathcal{A}$-SW in generative modeling is a good effect; however, it might not be guaranteed in other applications.
>
> **Q21**: Add a more extensive discussion on the amortization gap to better motivate the proposed amortized models and the training procedure.
>
> **A21**: We have added a paragraph in Appendix G for discussing the limitations of our work including the lack of understanding of the amortization gap of proposed amortized models.
>
>
> **Q22**:: Typos…
>
> **A22**: We have fixed all the mentioned typos in blue color for the revision. We really appreciate your feedback.

---

> ### Author Response · Authors · 2022-08-09
> **Look forward to your feedback.**
>
> Dear Reviewer hTkP,
>
> We have addressed your concerns in our responses. Given that the discussion deadline is only a few hours from now and you are the only one that gives a negative score on our paper, we would like to hear your feedback. Please feel free to raise questions if you have other concerns.
>
> Best regards,
>
> Authors

---

### Official Review · Reviewer_nJDk · 2022-07-10

**Rating:** 6
**Confidence:** 4
**Soundness:** 3 good
**Presentation:** 2 fair
**Contribution:** 3 good

**Summary:**

In the paper, the authors investigate the application of amortized optimization procedures to find the most distinguishing direction that can be used to calculate the sliced Wasserstein distance between two empirical latent distributions. This leads to the construction of three variants of the amortized sliced Wasserstein model: linear amortized ($\mathcal{LA}$-SW), generalized linear amortized ($\mathcal{GA}$-SW), and non-linear amortized ($\mathcal{NA}$-SW). The authors conduct experiments on CIFAR10, CelebA(-HQ), and  STL10 datasets, which prove the performance of their methods compared to SNGAN and (max-)SW models. They also analyze the computational complexity of the proposed solutions.

**Questions:**

My final opinion may depend on the response to issues (1) and (2).

(1)  I think the presentation (including the English of the paper) needs some improvements. For example, the authors quite often use simplified phrases such as (amortized, sliced, max) Wasserstein, to name the models, distances, or losses (without adding these words, what I find inappropriate), e.g.:

l. 43: Sliced Wasserstein is defined $\rightarrow$ Sliced Wasserstein distance is defined,

l. 69: named amortized sliced Wasserstein $\rightarrow$ named amortized sliced Wasserstein losses,

l. 141: mini-batch max-sliced Wasserstein $\rightarrow$ mini-batch max-sliced Wasserstein loss,

l. 222-223: leads to three corresponding amortized sliced Wasserstein $\rightarrow$ leads to three corresponding amortized sliced Wasserstein models.

(2) Do the sets of vectors listed in l. 192, 202, and 217 indeed form bases of linear subspaces in $\mathbb{R}^d$, or do only span these spaces (without independence), and why does $w_0$ appear only one time? Moreover, what do you exactly mean by writing "one-one "reshape" mapping"?

(3) Minor issues:

l. 27: $\phi$ is a parameter of NN (not NN),

l. 23 and 30: $\mathcal{D}(\cdot,\cdot)$ denotes an arbitrary discrepancy or an OT distance?,

l. 38: what is $d$?

l. 38: needs precising: I hope $m$ means the size of the sum of supports of two mini-batch measures,

l. 91-92:  a language issue in this sentence ("as the product measure which has the support is $m$ random variables follows $\mu$"?),

l. 127: huge/large,e.g., $\rightarrow$ huge/large, e.g., (missing gap),

l. 133: samples $\rightarrow$ sample,

l. 172 and 184: equation (5)/5 $\rightarrow$ Equation (5),

l. 174: measure $\rightarrow$ measures,

l. 213: the one-one mapping $\rightarrow$ the one-one "reshape" mapping,

l. 235: $\ldots$ $\rightarrow$ $, \ldots,$.

**Limitations:**

I would recommend including a separate paragraph to describe limitations (as it was done for potential impact, see Appendix F).

**Strengths And Weaknesses:**

The proposed method seems to be novel, well-motivated, and with sufficient theoretical background. Also, the experimental results show significant improvement over the SOTA techniques. Hence, I think this paper may be interesting for the ML community. However, I found some presentation issues (see below for the details).

---

> ### Author Response · Authors · 2022-08-01
> **Response to Reviewer nJDk**
>
> **Q8**: I think the presentation (including the English of the paper) needs some improvements. For example, the authors quite often use simplified phrases such as (amortized, sliced, max) Wasserstein, to name the models, distances, or losses (without adding these words, what I find inappropriate),
>
> **A8**: Thank you for your comments. We have revised the paper based on your suggestion. The modifications are written in blue color in the revision.
>
> **Q9**: Do the sets of vectors listed in l. 192, 202, and 217 indeed form bases of linear subspaces in  Rd, or do only span these spaces (without independence), and why does w0 appear only one time? Moreover, what do you exactly mean by writing "one-one "reshape" mapping"?
>
> **A9**: The set of vectors in l. 192, 202, and 217 are not guaranteed to be independent since they are subsampled from the data; hence, the rank of the linear subspace is less or equal than $2m+1$ where $m$ is the number of samples in mini-batches.  The reason why $w_0$ appears only once is that it is the vector of biases of the linear model. The one-one reshape mapping is the mapping that maps vectors of size $dn$ to a matrix of size $d\times n$. This mapping is for a rigorous definition of changing a vector into a matrix.
>
>
> **Q10**: $D$ denotes an arbitrary discrepancy or an OT distance?
>
> **A10**: Yes, we use $D(.,.)$ to denote an arbitrary discrepancy or an OT distance. The estimator in Equation 2 is normally known as the minimum distance estimator [R.3, R.4] while the estimator based on Equation 3 is known as the minimum expected distance estimator [R.3, R.4].
>
> [R.3] On parameter estimation with the Wasserstein distance, Bernton et al.
>
> [R.4] Asymptotic Guarantees for Learning Generative Models with the Sliced-Wasserstein Distance, Nadjahi et al
>
> **Q11**: what is d?
>
> **A11**:  $d$ is the dimension of supports of probability measures as mentioned in the Notation paragraph at the beginning of page 2.
>
> **Q12**: needs precising: I hope m means the size of the sum of supports of two mini-batch measures
>
> **A12**: It is correct, $m$ is the number of supports of two mini-batch measures.
>
> **Q13**: I would recommend including a separate paragraph to describe the limitations
>
> **A13**: We have added a paragraph for discussing the limitations of our proposed methods in Appendix G. In summary, we have not been able to investigate the amortization gaps of the proposed amortized models since the connection of the optima of Max-SW to the supports of two probability measures has not been well-understand yet. Moreover, the design of amortized models requires more engineering to achieve better performance since there is no inductive bias for designing them at the moment. The hardness in designing amortized models is that we need to trade-off between performance and computational efficiency. We will leave these questions to future work.

---

> > ### Comment · Reviewer_nJDk · 2022-08-05
> > **Response to Authors**
> >
> > Thank you very much for your response. I keep my (positive) rating unchanged. Let me refer to some points.
> >
> > **A9**. If independence is not guaranteed, these sets should not be formally called "bases" of linear spaces (recall that a basis of a linear space is a set of linearly independent vectors that spans the space). You can simply use the names "sets" or "collections". Also, thank you for the note on $w_0$, but this is not what I asked for. I wanted to know why $w_0$ was not an element of the spanning sets in Definitions 4 and 5 (which, as I supposed, was wrong). However, I see that in the revised version $w_0$ appears in each spanning set, so now it's ok.
> >
> > **A10**. I wanted to suggest not using the same notation for various objects. I find a discrepancy between distributions a more general notion than a distance/metric between distributions (e.g, OT distance).
> >
> > **A11**. Ok, but I suggest that, for the reader's convenience, it should be written before the first use (in l. 38).
> >
> > **A12**. By definition, the support of a probability measure is the largest (closed) set consisting of points for which every open neighborhood has a positive measure.

---

> > > ### Author Response · Authors · 2022-08-06
> > > **Response to Reviewer nJDk**
> > >
> > > Dear Reviewer nJDk,
> > >
> > > Thank you for your feedback. We will revise our paper based on your comments. Please feel free to raise questions if you want us to clarify anything about our work.
> > >
> > > Best regards,

---

### Official Review · Reviewer_mdhx · 2022-07-12

**Rating:** 8
**Confidence:** 3
**Soundness:** 4 excellent
**Presentation:** 4 excellent
**Contribution:** 4 excellent

**Summary:**

The presented draft proposes a method for estimating the Max-SW by learning the projection leading to the largest possible SW metric.

**Questions:**

Could the method be extended to “top-k SW” instead of max-SW?
Orthogonality could be enforced by a orthogonality loss via a Lagrangian.


**Limitations:**

The method requires additional memory and model evaluations but this is extensively discussed in the presented draft.

In practice with large models, i.e., in the demonstrated experiments, the computational cost of the Wasserstein loss is often small, which means that the utility of the method is limited in some applications. However, due to the favorable asymptotical complexity of the method, it allows learning in new settings which were previously impossible or hardly feasible. Further, even in the demonstrated cases the method achieve superior performance.

What I would have liked to see would be using SW and Max-SW with even larger L / T. It would be interesting to see at which point the methods break even, even if the baselines take a very long time.
It would strengthen the paper even further, and with such an evaluation, I would consider raising my score.

**Strengths And Weaknesses:**

### Strengths
Generative modeling is an important subject in machine learning.

The paper is exceptionally well written and simple to understand.
The proposed method is simple yet very powerful as it simplifies computation for Wasserstein generative models while improving model performance.

The background is very well explained. The method section is easy to understand.

The empirical evaluation is extensive, covering four data sets of different scales, and the results are impressive.

I greatly appreciate the submission of the code, which is well structured.


### Weaknesses

#### Miscellaneous Issues
* The presented paper uses a different font / font size or line distance. As the font / line distance is increased, I will not consider it for the review score, but it has be fixed for the final version.

#### Typos
* 35: there have been remained certain problems
* 85: we make some conclusion. Suggestions: we provide a conclusion.
* 86: preposition: defer … in -> defer … to
* 174: measure -> measures
* 223: leads to three corresponding amortized sliced Wasserstein [a word seems to be missing here], …

---

> ### Author Response · Authors · 2022-08-01
> **Response to Reviewer mdhx**
>
> **Q5**: Typos, fonts, and sizes…
>
> **A5**: Thank you for your comments. We have fixed all these typos and errors in blue color in the revision based on your suggestions.
>
>
> **Q6**: Could the method be extended to “top-k SW” instead of max-SW? Orthogonality could be enforced by an orthogonality loss via a Lagrangian.
>
> **A6**: In the revision, we introduce Amortized Projected Robust Wasserstein ($\mathcal{A}$-PRW) in Appendix C which is the application of amortized models into Projected Robust Wasserstein (PRW)  [2]. PRW is the generalization of Max-SW that finds the best orthonormal matrix $U \in \mathbb{R}^{d\times k}$ with $k>1$ that can maximize the projected Wasserstein distance between projected $k$ dimensional measures. Similar to Max-SW, PRW can be solved by using projected gradient descent with the projection operator be the QR decomposition. We refer the reviewer to Appendix C in the revision for more detail. By modifying our proposed amortized model slightly, we can derive three variants of  $\mathcal{A}$-PRW including $\mathcal{LA}$-PRW, $\mathcal{GA}$-PRW, and $\mathcal{NA}$-PRW. We again conduct experiments on generative models to compare $\mathcal{A}$-PRW variants with PRW with $k\in $ \{2,4,16\}  on CIFAR10 dataset. The result is given in Table 4 in Appendix E and the detailed settings are also given in Appendix E. From the table, we observe that $\mathcal{A}$-PRW variants give both lower FID scores and IS scores than PRW for all choices of $k$. This strengthens the application of amortized optimization in finding the best subspace and projecting directions for comparing probability measures.
>
> **Q7**: What I would have liked to see would be using SW and Max-SW with even larger L / T. It would be interesting to see at which point the methods break even, even if the baselines take a very long time. It would strengthen the paper even further, and with such an evaluation, I would consider raising my score.
>
> **A7**: Due to the limitation of time and our hardware, we cannot run experiments with larger $L/T$ since they either exceed our GPUs’ memory or consume a lot of time.  We will try to optimize the memory consumption and report the result in the discussion when it is available.

---

> ### Author Response · Authors · 2022-08-06
> **Look forward to your feedback.**
>
> Dear Reviewer mdhx,
>
> We have addressed your concerns in our responses. We would like to hear your feedback. Please feel free to raise questions if you have other concerns.
>
> Best regards,
>
> Authors

---

> > ### Comment · Reviewer_mdhx · 2022-08-08
> > **Thanks**
> >
> > Thank you for your clear response. As it is already a strong accept, I will keep my score.
> >
> > I wanted to shortly note that Fig. 1 and Table 1 are slightly to large (which obviously does not impact my decision).

---

> > > ### Author Response · Authors · 2022-08-08
> > > **Response to Reviewer mdhx**
> > >
> > > Dear Reviewer mdhx,
> > >
> > > Thank you for your time and feedback. We will revise the size of Fig. 1 and Tabl 1 in the revision.
> > >
> > > Best regards,
> > >
> > > Authors

---

### Official Review · Reviewer_3bjP · 2022-07-17

**Rating:** 6
**Confidence:** 3
**Soundness:** 3 good
**Presentation:** 3 good
**Contribution:** 2 fair

**Summary:**

This work proposes to use amortized optimization to predict the optimal projection direction which can be used for learning Wasserstein generative models. The authors derived a family of amortized models which can be used for the framework. Empirical results show that the proposed approach is able to achieve good performance on benchmark datasets.

**Questions:**

1. Proposition 1 mentions that $A-SW(\mu, \nu)=0$ does not imply $\mu=\nu$. Would the proposed method still guarantee convergence to the right distribution?
2. As mentioned in Line 233, the optimization forms a min-max problem. Does it cause training instabilities?
3. Does the batch size affect the performance? In other words, does the proposed approach require significantly larger batch sizes to achieve good performance?


**Limitations:**

The authors have not adequately addressed the limitations and potential negative societal impact of their work. It would be good to add more discussions.

**Strengths And Weaknesses:**

**Strength**
1. The paper is relatively well-written.
2. The experiments are detailed.
3. The proposed approach could be interesting for the community. However, learning the projection direction also seems straightforward.

**Weakness**
1. The technical novelty could be limited. The proposed approach seems like a direct application of amortized inference to sliced Wasserstein generative models. There are also no extra theoretical insights/analyses provided.
2. The amortized model considered in the paper could be restrictive. It would be good to perform theoretical analyses on how powerful the model family is.
3. Empirical results are good but not impressive. The improvements in Table 1 are marginal. In Table 2, the memory cost for the proposed approach is also larger than the baselines.

---

> ### Author Response · Authors · 2022-08-01
> **Response to Reviewer 3bjP - Part 1**
>
> **Q1**: The proposed approach seems like a direct application of amortized inference to sliced Wasserstein generative models. There are also no extra theoretical insights/analyses provided. The amortized model considered in the paper could be restrictive. It would be good to perform theoretical analyses on how powerful the model family is.
>
> **A1**: Thanks for your comment. We would like to recall that this is the first work that connects amortized optimization to the sliced Wasserstein literature, hence, the design of amortized models are original. To our best knowledge, the understanding of the optimality of max-sliced Wasserstein has not been established yet. Therefore, understanding the amortization gap and designing good amortized models have still remained to be open questions. We believe that it will require more effort in engineering to find the best design in the case of mini-batch max sliced Wasserstein loss. We leave this investigation to future work.
>
> According to our insight of designing amortized models, for the linear model, the assumption is that the optimal projecting direction lies on the subspace that is spanned by the basis \{$x_1,\ldots,x_m,y_1,\ldots,y_m,w_0$\} where $X=(x_1,\ldots,x_m)$ and $Y=(y_1,\ldots,y_m) $ are supports of two mini-batch measures, and $w_0$ is the vector of biases of the linear model. Similarly, the generalized linear model assumes that the optimal projecting direction lies in the subspace \{$x_1’,\ldots,x_m’,y_1’,\ldots,y_m’,w_0$\} where $X’=(x_1’,\ldots,x_m’) = g_{\psi_1}(X)$ and $Y’=(y_1’,\ldots,y_m’)= g_{\psi_1}(Y)$ for some non-linear link function $ g_{\psi_1}(.)$. The above two amortized models are motivated by the fundamental literature on generalized linear models. Moreover, we would like to recall that we can use several architectures of neural networks for the amortized model. In the paper, our non-linear amortized model can be seen as a two-layer MLP. However, a more powerful amortized model leads to higher computational complexity. As we focus on the efficiency of the model instead of the performance, we only design the amortized models to be as simple as possible.
>
> From the experimental result, we observe that our proposed models have a good computational speed and also provide good performance.  We also observe the trade-off between training efficiency and performance of amortized models.  In particular, the non-linear amortized model provides the best result on CelebA dataset; however, its computational time and memory are worse than the linear amortized model (see Table 2 in the main text). Again, this is the trade-off that exists in almost all deep learning applications.
>
> **Q2**: Empirical results are good but not impressive. The improvements in Table 1 are marginal. In Table 2, the memory cost for the proposed approach is also larger than the baselines.
>
> **A2**: Thanks for your comment. We would like to remark that the main comparison that we want to make is between $\mathcal{A}$-SW, SW, and Max-SW. The linear version of $\mathcal{A}$-SW -$\mathcal{ LA}$-SW is better than both SW and Max-SW in terms of performance, memory, and speed. All sliced Wasserstein variants have higher memory than SNGAN; however, their performance is significantly better than that of SNGAN. To our best knowledge,  we have achieved the best performance of SNGAN architecture in generative modeling. As a reference, we would like to refer the reviewer to Table 3 in [R.1] for the current best FID scores of SNGAN architecture. In our paper, we have achieved a lower (better) FID score. Therefore, we could say that our experimental result is enough to show that ASW variants help to learn better generative models than SW and prior approaches.
>
> Moreover, our main contribution is connecting the literature on sliced Wasserstein and amortized optimization. $\mathcal{A}$-SW can be used in several other deep learning applications that involve comparing probability measures, such as domain adaptation, approximate inference, adversarial attacks, and so on. Therefore, we believe that the impact of $\mathcal{A}$-SW is promising.
>
> [R.1] Exploiting Chain Rule and Bayes’ Theorem to Compare Probability Distributions, Huangjie Zheng et al, NeuRIPS 2021.

---

> ### Author Response · Authors · 2022-08-01
> **Response to Reviewer 3bjP - Part 2**
>
> **Q3**: Would the proposed method still guarantee convergence to the right distribution?
>
> **A3**: Thanks for your question. To our best knowledge, there is no deep generative model that can guarantee convergence in the current literature. The main reason for the lacking of theoretical analysis is due to the usage of minibatch in deep learning applications [1]. The minibatch variants of Wasserstein are not a proper metric in the probability spaces [R. 2], e.g., minibatch with Wasserstein distance, Sinkhorn divergence, sliced Wasserstein, max-sliced Wasserstein, and amortized sliced Wasserstein for deep generative models. Despite lacking metricity, the empirical results indicate that using (amortized) sliced Wasserstein still produces a better result than the conventional GAN training. The improvement in terms of performance indicates that the proposed method may potentially guarantee a better estimation of the underlying distribution than GAN. We leave a rigorous investigation of that conjecture for future work.
>
> [R. 2] Learning with minibatch Wasserstein : asymptotic and gradient properties, Fatras et al
>
> **Q4**: As mentioned in Line 233, the optimization forms a min-max problem. Does it cause training instabilities?
>
> **A4**: Due to limitations of time and hardware, we can only run SW ($L \in $\{100,1000,10000\}), Max-SW (slice learning rate $ \eta_2 \in $\{0.001,0.01\}  and $ T_2 \in$ \{1,10,100\}), $\mathcal{LA}$-SW, $\mathcal{GA}$-SW, and $\mathcal{NA}$-SW five different times on CIFAR10. We evaluate the FID scores and IS scores on only the last epoch to speed up the experiments. We have updated the results with error bars in Table 1 in the revision.  From the table, we observe that Namely, $\mathcal{LA}$-SW and $\mathcal{NA}$-SW give the best FID score and IS score respectively on CIFAR10. Moreover, we observe that the results of $\mathcal{A}$-SW losses have relatively small error bars. Overall, the current setting of architectures of neural networks and training procedures are stable for $\mathcal{A}$-SW losses.
>
> **Q5**: Does the batch size affect the performance? In other words, does the proposed approach require significantly larger batch sizes to achieve good performance?
>
> **A5**: The batch size does not affect the relative comparison between $\mathcal{A}$-SW, mini-batch SW, and mini-batch Max-SW. However, a bigger batch size leads to a better estimation of mini-batch distribution to the original measure. Namely, the sample complexity of sliced Wasserstein is $O(m^{-1/2})$ where $m$ is the size of mini-batches. However, a better estimation might not lead to a better FID score and a better IS score since they have different favors in measuring discrepancy between distributions. Due to the limitation of time and hardware, we have not been able to run additional experiments on changing the batch size. We will add the result to the discussion when it is available.
>
> **Q6**: The authors have not adequately addressed the limitations and potential negative social impact of their work. It would be good to add more discussions.
>
> **A6**: Thank you for your feedback. We have added more discussions on the potential negative social impact and limitations of our proposed methods in Appendix G. In summary, amortized sliced Wasserstein losses can be applied to various applications such as generative models, domain adaptation, and approximate inference, adversarial attack, and so on. Due to its widely used potential, it can be used as a component in some applications that do not have a good purpose. For example, some examples are creating images of people without permission, attacking machine learning systems, and so on.

---

> > ### Author Response · Authors · 2022-08-07
> > **Ablation study of changing batch size $m$**
> >
> > Dear Reviewer 3bjP,
> >
> > we have run experiments on CIFAR10 with the batch size $m=32$, $m=64$ in addition to $m=128$ in the paper. In more detail, we run SW ($L=1000$), Max-SW ($\eta_2=0.01$, $T_2 =10$), LA-SW, GA-SW, and NA-SW three times. We observe that increasing batch size leads $m$ to better FID scores and IS scores for all methods. This is consistent with the theoretical sample complexity of sliced Wasserstein. For all choices of $m$, A-SW variants are better than SW and Max-SW. When the mini-batch size is small e.g, 32 and 64, GA-SW and NA-SW are better than LA-SW. The reason might be the non-linearity in GA-SW and NA-SW is better than the linearity in LA-SW when having limited data in mini-batches. The FID scores and the IS scores are given below.
> >
> > |Method |FID  | IS|
> > --- | --- | ---|
> > |SW m=32 |19.56 $\pm$ 0.41|7.80 $\pm$ 0.03|
> > |Max-SW m=32 |36.11 $\pm$ 1.45|6.45 $\pm$ 0.37|
> > |LA-SW m=32|19.45 $\pm$ 0.05|7.91$\pm$ 0.04|
> > |GA-SW m=32|**17.53 $\pm$ 0.81**|**7.95 $\pm$ 0.08**|
> > |NA-SW m=32|18.84 $\pm$ 1.02|7.89 $\pm$ 0.11 |
> > --- | --- | ---|
> > |SW m=64 |16.02 $\pm$ 0.87|8.01 $\pm$ 0.11|
> > |Max-SW m=64 |34.33 $\pm$ 1.22|6.57 $\pm$ 0.25|
> > |LA-SW m=64|15.99 $\pm$ 0.12|7.98$\pm$ 0.08|
> > |GA-SW m=64|15.22 $\pm$ 0.31|**8.09 $\pm$ 0.02**|
> > |NA-SW m=64|**15.15$\pm$ 0.58**|8.08 $\pm$ 0.10 |
> > --- | --- | ---|
> > |SW m=128 |14.25 $\pm$ 0.80 |8.12 $\pm$ 0.07|
> > |Max-SW m=128 |31.33 $\pm$ 3.02|6.67 $\pm$ 0.37|
> > |LA-SW m=128|**13.21 $\pm$ 0.69** |8.19 $\pm$ 0.03|
> > |GA-SW m=128|13.64 $\pm$ 0.11|8.22 $\pm$ 0.11|
> > |NA-SW m=128|14.22 $\pm$ 0.51|**8.29 $\pm$ 0.08** |

---

> ### Author Response · Authors · 2022-08-08
> **Look forward to your feedback.**
>
> Dear Reviewer 3bjP,
>
> We have addressed your concerns in our responses. Given that the discussion deadline is approaching, we would like to hear your feedback. Please feel free to raise questions if you have other concerns.
>
> Best regards,
>
> Authors

---

### Author Response · Authors · 2022-08-01
**Summary of the revision**

Dear Reviewers and Chairs,

We would like to thank the reviewers for their time and feedback. We have answered all questions of the reviewers in the corresponding discussions. Moreover, we have also included the following results (written in blue color) in our revision:

1. We apply amortized optimization to mini-batch projected robust Wasserstein (PRW) to obtain Amortized projected robust Wasserstein ($\mathcal{A}$-PRW) . With a slight modification of amortized models, we introduce linear Amortized projected robust Wasserstein $\mathcal{LA}$-PRW, generalized linear Amortized projected robust Wasserstein $\mathcal{GA}$-PRW, and non-linear Amortized projected robust Wasserstein $\mathcal{NA}$-PRW. We refer reviewers to Appendix C for detailed definitions. We conduct experiments on generative models on CIFAR10 to compare $\mathcal{A}$-PRW to mini-batch PRW. Overall, we observe that $\mathcal{A}$-PRW gives a better FID score and IS score than PRW.

2. We run mini-batch SW (L=100,1000,10000), mini-batch Max-SW ($\eta_2 \in$ \{0.001,0.01\} and $T_2$ in \{1,10,100\}), $\mathcal{LA}$-SW, $\mathcal{GA}$-SW, and $\mathcal{NA}$-SW five different times on CIFAR10. We have updated the result in Table 1 in the main text. We still observe that $\mathcal{LA}$-SW and $\mathcal{NA}$-SW give the best FID score and IS score respectively.

3. We have added a paragraph in Appendix G for discussing the limitations of our proposed methods.

4. We have also submitted the code for the new experiments.

5. We have fixed typos and revised the writing based on the suggestions of reviewers in blue color.

We are looking forward to your feedback.

Best regards,

Authors

---

### Meta-Review · Area_Chair_ZWgs · 2022-08-26

**Recommendation:** Accept
**Confidence:** Less certain

**Metareview:**

During the author-reviewer discussions, the authors have addressed most of the concerns raised by the reviewers, leading to original scores being raised. During the reviewer discussions, the disagreement among reviewers about the demonstration of computational benefits was discussed. At this point, the merits of the paper, including the originality of its contribution and the sufficient experimental validation, outweigh the doubts remaining with one of the reviewers. Therefore, the recommendation is to accept this submission.

I would like to thank the authors and reviewers for engaging in discussions.


**Award:**

No

---

### Decision · Program_Chairs · 2022-09-14

Accept